# Unveiling the Potential of Robustness in Selecting Conditional Average Treatment Effect Estimators

**Yiyan Huang** *
The Hong Kong Polytechnic University
yiyhuang3-c@my.cityu.edu.hk

**Cheuk Hang Leung** *
City University of Hong Kong
chleung87@cityu.edu.hk

**Siyi Wang**
City University of Hong Kong
siyi.wang@my.cityu.edu.hk

**Yijun Li**
City University of Hong Kong
yijunli5-c@my.cityu.edu.hk

**Qi Wu** †
City University of Hong Kong
qi.wu@cityu.edu.hk

## Abstract

The growing demand for personalized decision-making has led to a surge of interest in estimating the Conditional Average Treatment Effect (CATE). Various types of CATE estimators have been developed with advancements in machine learning and causal inference. However, selecting the desirable CATE estimator through a conventional model validation procedure remains impractical due to the absence of counterfactual outcomes in observational data. Existing approaches for CATE estimator selection, such as plug-in and pseudo-outcome metrics, face two challenges. First, they must determine the metric form and the underlying machine learning models for fitting nuisance parameters (e.g., outcome function, propensity function, and plug-in learner). Second, they lack a specific focus on selecting a robust CATE estimator. To address these challenges, this paper introduces a Distributionally Robust Metric (DRM) for CATE estimator selection. The proposed DRM is nuisance-free, eliminating the need to fit models for nuisance parameters, and it effectively prioritizes the selection of a distributionally robust CATE estimator. The experimental results validate the effectiveness of the DRM method in selecting CATE estimators that are robust to the distribution shift incurred by covariate shift and hidden confounders.

## 1 Introduction

The escalating demand for decision-making has sparked an increasing interest in *Causal Inference* across various research domains, such as economics [23, 13, 43, 1], statistics [70, 49, 26, 41], healthcare [78, 27, 61, 9, 42], and financial application [11, 15, 37, 21, 35, 24, 50]. The primary goal in personalized decision-making is to quantify the causal effect of a specific treatment (or policy/intervention) on the target outcome, and understanding such causal effects is closely connected with identifying the *Conditional Average Treatment Effect (CATE)*. In observational studies, identifying the CATE faces a significant and fundamental challenge: the absence of *counterfactual* knowledge. According to Rubin Causal Model [64], the CATE is determined by comparing *potential outcomes*

---

*The co-first authors.
†Corresponding author.

38th Conference on Neural Information Processing Systems (NeurIPS 2024).

under different treatment assignments (i.e., treat and control) for a specific covariate. Nonetheless, in real-world applications, we can only observe the potential outcome under the actual treatment (i.e., *factual outcome*), while the potential outcome under the alternative treatment (i.e., *counterfactual outcome*) remains unobserved. The unavailability of the counterfactual outcome is widely recognized as the fundamental problem in causal inference [33], making it difficult to accurately determine the true value of the CATE.

The advancement of machine learning (ML) has opened up a promising opportunity to improve the CATE estimation from observational data. Several innovative CATE estimation approaches, such as meta-learners and causal ML models, have been proposed to tackle the fundamental challenge in causal inference and enhance the predictive accuracy of CATE estimates (as discussed in Section 3). Nevertheless, the emergence of various CATE estimation methods has brought forth a new question: ***Given multifarious options for CATE estimators, which should be chosen?*** In observational data, treatment is often non-random and propensity scores remain unknown. Conventional model validation procedures, unfortunately, are not suitable for CATE estimator selection in this case due to the absence of ground truth CATE labels. Therefore, exploring proper metrics for CATE estimator selection remains an essential yet challenging research topic in causal inference.

Recent research has emphasized the significance of model selection for CATE estimators, as highlighted in [66, 20, 53]. These works have proposed and summarized two types of criteria for CATE estimator selection, the *plug-in* metric $\mathcal{R}^{plug}_{\tilde{\tau}}(\hat{\tau})$ and the *pseudo-outcome* metric $\mathcal{R}^{pseudo}_{\tilde{Y}}(\hat{\tau})$:

$$\mathcal{R}^{plug}_{\tilde{\tau}}(\hat{\tau}) = \sqrt{\frac{1}{n}\sum_{i=1}^{n}(\hat{\tau}(X_i) - \tilde{\tau}(X_i))^2}, \quad \mathcal{R}^{pseudo}_{\tilde{Y}}(\hat{\tau}) = \sqrt{\frac{1}{n}\sum_{i=1}^{n}(\hat{\tau}(X_i) - \tilde{Y}_i)^2}. \quad (1)$$

One can establish a plug-in estimator $\tilde{\tau}$ or construct a pseudo-outcome estimator $\tilde{Y}$ using the validation data to select CATE estimator $\hat{\tau}$. The previous studies [66, 20, 53] have shown that these metrics offer some assistance in identifying well-performing CATE estimators. However, two additional challenges are still encountered in these two metrics.

***Challenge 1: How to determine the metric form and underlying ML models for nuisance parameters?*** As previously discussed, plug-in and pseudo-outcome metrics have various forms, and both of them rely on estimating nuisance parameters $\tilde{\eta}$ using ML algorithms such as linear models, tree-based models, etc. Plug-in metrics even need to fit an additional ML model for the plug-in learner $\tilde{\tau}$. However, selecting the suitable metric form and ML algorithms can be very difficult without the knowledge of true data generating process. Consequently, we might go round in circles as this challenge leads us back to the original estimator selection problem [20].

***Challenge 2: These metrics are not well-targeted for selecting a robust CATE estimator.*** In potential outcome framework [64], the factual distribution $P^F$ and the counterfactual distribution $P^{CF}$ for $t \in \{0,1\}$ can be defined as follows:

$$P^F := P(X, Y^t|T = t) = P(Y^t|X, T = t)P(X|T = t);$$
$$P^{CF} := P(X, Y^t|T = 1-t) = P(Y^t|X, T = 1-t)P(X|T = 1-t). \quad (2)$$

The above (2) reveals that the covariate shift $P(X|T = t) \neq P(X|T = 1-t)$ leads to a distribution shift between $P^F$ and $P^{CF}$ - and such distribution shift can be further exacerbated once the unconfoundedness assumption $P(Y^t|X, T = t) = P(Y^t|X, T = 1-t)$ is violated. It is widely recognized that ML models often struggle when the training and test data do not adhere to the same distribution. Therefore, it becomes essential to select a CATE estimator learned on $P^F$ that demonstrates robust performance to the counterfactual distribution $P^{CF}$. This need for robustness holds even greater significance than the pursuit of an ideal "stellar" estimator because striving for the perfect estimator can be futile in the absence of ground truth counterfactual labels.

**Contributions.** In this paper, we propose a Distributionally Robust Metric (DRM) for CATE estimator selection. The main contributions are summarized as follows: (1) The proposed DRM method is nuisance-free, eliminating the need to fit models for nuisance parameters (outcome function, propensity function, and plug-in learner). (2) The DRM method is designed to prioritize selecting a distributionally robust CATE estimator. (3) We provide a finite sample analysis of the proposed

distributionally robust value $\hat{\mathcal{V}}^t(\hat{\tau})$ for $t \in \{0, 1\}$, showing it decays to $\mathcal{V}^t(\hat{\tau})$ at a rate of $n^{-1/2}$. (4) Experimental results validate the effectiveness of the DRM method in selecting a CATE estimator that is robust to the distribution shift incurred by covariate shift and hidden confounders.

## 2 Background of CATE Estimator Selection

Suppose the observational data contain $n$ i.i.d. samples $\{(x_i, t_i, y_i)\}_{i=1}^n$, with the associated random variables being $\{(X_i, T_i, Y_i)\}_{i=1}^n$. For each unit $i$, $X_i \in \mathcal{X} \subset \mathbb{R}^d$ is $d$-dimensional covariates and $T_i \in \{0, 1\}$ is the binary treatment. Potential outcomes for treat ($T = 1$) and control ($T = 0$) are denoted by $Y^1, Y^0 \in \mathcal{Y} \subset \mathbb{R}$. The observed (factual) outcome is $Y = TY^1 + (1 - T)Y^0$. The propensity score [63] is defined as $\pi(x) := P(T = 1 \mid X = x)$. The conditional mean potential outcome surface is defined as $\mu_t(x) := \mathbb{E}[Y^t \mid X = x]$ for $t \in \{0, 1\}$. The true CATE is defined as

$$\tau_{true}(x) := \mathbb{E}[Y^1 - Y^0 \mid X = x] = \mu_1(x) - \mu_0(x).$$

Following the standard and necessary assumptions in potential outcome framework [64], we impose Assumption 2.1 that ensure treatment effects are identifiable.

**Assumption 2.1** (Consistency, Overlap, and Unconfoundedness)**.** Consistency: If the treatment is $t$, then the observed outcome $Y$ equals $Y^t$. Overlap: The propensity score is bounded away from 0 to 1, i.e., $0 < \pi(x) < 1, \forall x \in \mathcal{X}$. Unconfoundedness [3]: $Y^t \perp\!\!\!\perp T \mid X, \forall t \in \{0, 1\}$.

The goal of CATE estimator selection is to select the best CATE estimator, denoted by $\hat{\tau}_{best}$, from a set of $J$ candidate estimators $\{\hat{\tau}_1, \ldots, \hat{\tau}_J\}$:

$$\hat{\tau}_{best} = \operatorname*{arg\,min}_{\hat{\tau} \in \{\hat{\tau}_1, \ldots, \hat{\tau}_J\}} \mathcal{R}^{oracle}(\hat{\tau}), \quad \mathcal{R}^{oracle}(\hat{\tau}) := \sqrt{\frac{1}{n} \sum_{i=1}^n (\hat{\tau}(X_i) - \tau_{true}(X_i))^2}. \quad (3)$$

Here, $\mathcal{R}^{oracle}(\hat{\tau})$ is associated with $\mathbb{E}[(\hat{\tau}(X) - \tau_{true}(X))^2]$, known as the Precision of Estimating Heterogeneous Effects (PEHE) w.r.t. $\hat{\tau}$ [32, 68]. Note that $\mathcal{R}^{oracle}(\hat{\tau})$ cannot be employed to evaluate CATE estimators' performances in real applications as we do not have access to $\tau_{true}$. Previous studies have introduced plug-in and pseudo-outcome metrics to aid in CATE estimator selection, as shown in equation (1). Then, the CATE estimator $\hat{\tau}_{select}$ is selected on validation data by

$$\hat{\tau}_{select} = \operatorname*{arg\,min}_{\hat{\tau} \in \{\hat{\tau}_1, \ldots, \hat{\tau}_J\}} \mathcal{R}^{plug}_{\tilde{\tau}}(\hat{\tau}) \quad \text{or} \quad \hat{\tau}_{select} = \operatorname*{arg\,min}_{\hat{\tau} \in \{\hat{\tau}_1, \ldots, \hat{\tau}_J\}} \mathcal{R}^{pseudo}_{\tilde{Y}}(\hat{\tau}). \quad (4)$$

Notably, both the plug-in and pseudo-outcome metrics necessitate the fitting of nuisance parameters $\tilde{\eta}$ (e.g., $\tilde{\eta} = (\tilde{\mu}_1, \tilde{\mu}_0, \tilde{\pi})$) using off-the-shelf ML models. While some papers like [16] address the selection of nuisance parameters for Averate Treatment Effect (ATE) estimators, e.g., the doubly robust estimator [23, 13, 36], our paper focuses on the selection of CATE estimators rather than nuisance parameters. For the plug-in metric, $\tilde{\tau}$ can be constructed using any CATE estimator discussed in Appendix A.1, yielding metrics such as plug-T, plug-DR, etc. For the pseudo-outcome metric, $\tilde{Y}$ can be constructed using a specific formula discussed in Appendix A.2, yielding metrics such as pseudo-DR, pseudo-R, etc. The metrics based on the influence function [5] and the R-learner objective [55] are categorized into the pseudo-outcome metric. The categorization of plug-in and pseudo-outcome metrics maintains consistency with [20, 53].

## 3 Related Work

**CATE estimation.** Recent advancements in ML have emerged as powerful tools for estimating CATE from observational data, and researchers pay particular attention to *meta-learners* and *causal ML* models. Existing meta-learners mainly include traditional learners such as S-learner, T-learner, PS-learner, and IPW-learner, as well as new learners such as X-learner [47], U-learner [25, 55], DR-learner [41, 26], R-learner [55], and RA-learner [18]. The specific details of these meta-learners are stated in Appendix A.1. Additionally, some studies also focus on developing innovative causal ML

---

[3]Note that in the setting C of our experiments, the unconfoundedness assumption is violated, leading to misspecified nuisance parameters in CATE estimators and baseline selectors.

models for CATE estimation, such as Causal BART [30], Causal Forest [70, 8, 58], generative models like CEVAE [51] and GANITE [77], representation learning nets including SITE [76], TARNet [68], Dragonnet [69], FlexTENet [19], and HTCE [10], disentangled learning nets like D$^2$VD [44, 45], DeR-CFR [74], and DR-CFR [31], and representation balancing nets such as BNN [39], CFRNet [68], DKLITE [79], IGNITE [29], BWCFR [6], DRRB [35], and DIGNet [38]. Recent surveys [28, 75, 56] have also conducted a systematic review of various causal inference methods.

**CATE estimator selection.** Compared to the diverse range of CATE estimation methods, selecting CATE estimators has received limited attention in existing causal inference research. Current methods for selecting CATE estimators can be broadly classified into two main categories. **The first category**, which is also considered in this paper, involves using plug-in and pseudo-outcome methods to evaluate CATE estimators. These methods share two common characteristics: 1) Both methods require fitting ML models for nuisances (e.g., outcome function, propensity function, CATE function) on a validation set and then implementing the learned ML models in either the plug-in surrogate or the pseudo-outcome surrogate; 2) Both methods serve as surrogates for the expected error between the CATE estimator and the true CATE, i.e., $\mathcal{R}^{oracle}(\hat{\tau})$ in equation (3). The difference between the two methods is that the plug-in method directly approximates the true CATE function, where only covariate variables are involved, while the pseudo-outcome method typically constructs a specific formula incorporating covariates, treatment, and outcome variables. For example, the pseudo-DR proposed in [65] is constructed by the outcome predictors learned with representation balancing objective [68, 40]. Recent research [66, 20, 53] has conducted thorough empirical investigations into exploring these two methods for selecting CATE estimators. Their findings suggest that no single selection criterion can universally outperform others in all scenarios in the task of selecting CATE estimators. More details of the two selection methods are stated in Appendix A.2. **The second category** considers leveraging the data generating process (DGP) to generate synthetic data with the known true CATE function, allowing the validation of CATE estimators' performance on this synthetic data. For example, authors in [2] find that placebo and structured empirical Monte Carlo methods are helpful for estimator selection under some restrictive conditions. In addition, researchers in [67, 7, 59] focus on training generative models to enforce the generated data to approximate the distribution of the observed data. However, the DGP-based method still faces some limitations in CATE estimator selection: 1) it only guarantees the resemblance of the generated data to the factual distribution, without considering the counterfactual distribution; and ii) there is a potential risk of the method favoring estimators that closely resemble the generative models [17].

## 4 The Distributionally Robust Metric

In this section, we introduce the Distributionally Robust Metric (DRM) for CATE estimator selection. First, we capture the uncertainty in PEHE in a distributionally robust manner (Section 4.1). We then establish the DRM based on the distributionally robust value of PEHE (Section 4.2).

### 4.1 Capturing the Uncertainty in PEHE

**Proposition 4.1.** *The PEHE w.r.t. the CATE estimator $\hat{\tau}$ can be decomposed as follows:*

$$\mathbb{E}[(\hat{\tau}(X) - \tau_{true}(X))^2] = \mathbb{E}[\hat{\tau}(X)^2] + 2\mathbb{E}[\hat{\tau}(X)Y^0] + 2\mathbb{E}[-\hat{\tau}(X)Y^1] + \zeta, \quad (5)$$

*where $\zeta = \mathbb{E}[(\mu_1(X) - \mu_0(X))^2]$. The proof is deferred to Appendix B.1.*

Proposition 4.1 indicates that the PEHE is equal to four terms, where $\mathbb{E}[\hat{\tau}(X)^2]$, $\mathbb{E}[\hat{\tau}(X)Y^0]$, and $\mathbb{E}[-\hat{\tau}(X)Y^1]$ depend on $\hat{\tau}$, while $\zeta$ is a constant that is independent of $\hat{\tau}$. The term $\mathbb{E}[\hat{\tau}(X)Y^t]$ for $t \in \{0, 1\}$ can be further decomposed as follows:

$$\mathbb{E}[\hat{\tau}(X)Y^t] = \underbrace{\mathbb{E}[\hat{\tau}(X)Y^t|T = t]}_{\text{(a) Empirically computable}} P(T = t) + \underbrace{\mathbb{E}[\hat{\tau}(X)Y^t|T = 1 - t]}_{\text{(b) Empirically uncomputable}} P(T = 1 - t). \quad (6)$$

Equation (6a) can be computed empirically since the potential outcome $Y^t$ is observable in the group of $T = t$. However, equation (6b) is empirically uncomputable due to the unavailability of $Y^t$ in the group of $T = 1 - t$. The unknown term $\mathbb{E}[\hat{\tau}(X)Y^t|T = 1 - t]$ therefore determines the uncertainty in PEHE. To capture such an uncertainty, we therefore establish distributionally robust values for $\mathbb{E}[\hat{\tau}(X)Y^0|T = 1]$ and $\mathbb{E}[-\hat{\tau}(X)Y^1|T = 0]$ based on a Kullback-Leibler (KL) ambiguity set.

**Definition 4.2** (KL ambiguity set). Given two distributions $Q$ and $P$ and the ambiguity radius $\epsilon > 0$. The KL ambiguity (uncertainty) set $\mathcal{B}_\epsilon(P)$ is defined as

$$\mathcal{B}_\epsilon(P) := \{Q : D_{KL}(Q\|P) \leq \epsilon\}, \quad \text{where } D_{KL}(Q\|P) = \int_{\mathcal{X}} q(x) \log \frac{q(x)}{p(x)} dx. \qquad (7)$$

Here, $D_{KL}(Q\|P)$ denotes the KL divergence of some arbitrary distribution $Q$ from the reference distribution $P$. Now we define the distribution of $(X, Y^0, Y^1)$ in the treated and controlled groups as

$$P_T := P(X, Y^0, Y^1 | T = 1); \ P_C := P(X, Y^0, Y^1 | T = 0). \qquad (8)$$

By setting an adequately large ambiguity radius in Definition 4.2, the following inequalities hold for $\mathbb{E}[\hat{\tau}(X)Y^0 | T = 1] = \mathbb{E}^{P_T}[\hat{\tau}(X)Y^0]$ and $\mathbb{E}[-\hat{\tau}(X)Y^1 | T = 0] = \mathbb{E}^{P_C}[-\hat{\tau}(X)Y^1]$:

$$\mathbb{E}[\hat{\tau}(X)Y^0 | T = 1] = \mathbb{E}^{P_T}[\hat{\tau}(X)Y^0] \leq \sup_{Q \in B_{\epsilon_0}(P_C)} \mathbb{E}^Q[\hat{\tau}(X)Y^0] =: \mathcal{V}^0(\hat{\tau});$$

$$\mathbb{E}[-\hat{\tau}(X)Y^1 | T = 0] = \mathbb{E}^{P_C}[-\hat{\tau}(X)Y^1] \leq \sup_{Q \in B_{\epsilon_1}(P_T)} \mathbb{E}^Q[-\hat{\tau}(X)Y^1] =: \mathcal{V}^1(\hat{\tau}). \qquad (9)$$

To provide a clearer understanding, let us consider the example of $\mathbb{E}^{P_T}[\hat{\tau}(X)Y^0]$. Since the term $\mathbb{E}[\hat{\tau}(X)Y^0]$ is computable on its factual distribution $P_C$ but uncomputable on its counterfactual distribution $P_T$, we can construct an ambiguity set centered around the distribution $P_C$ such that it is large enough to contain the distribution $P_T$. By doing so, we can capture the uncertainty of $\mathbb{E}^{P_T}[\hat{\tau}(X)Y^0]$ w.r.t. $\hat{\tau}$. In other words, the value of the uncomputable quantity $\mathbb{E}^{P_T}[\hat{\tau}(X)Y^0]$ will be **at most** $\mathcal{V}^0(\hat{\tau})$. Similarly, the value of the uncomputable quantity $\mathbb{E}^{P_C}[-\hat{\tau}(X)Y^1]$ will be **at most** $\mathcal{V}^1(\hat{\tau})$. Obviously, the uncertainty in PEHE will be larger if the distribution shift between factual and counterfactual distribution is severer. Consequently, we can obtain the distributionally robust value of PEHE in Corollary 4.3, which measures the uncertainty in PEHE.

**Corollary 4.3.** *Let $\mathcal{V}^0(\hat{\tau})$ and $\mathcal{V}^1(\hat{\tau})$ be the quantities defined in equation (9), $\zeta$ be the constant given in Proposition 4.1, $u_1 := P(T = 1)$, and $u_0 = 1 - u_1 = P(T = 0)$. The distributionally robust value of PEHE w.r.t. $\hat{\tau}$ is defined as $\mathcal{V}_{PEHE}(\hat{\tau})$ such that*

$$\mathbb{E}[(\hat{\tau}(X) - \tau_{true}(X))^2] \leq \mathcal{V}_{PEHE}(\hat{\tau})$$
$$= \mathbb{E}[\hat{\tau}(X)^2] + 2\left(u_0 \mathbb{E}^{P_C}[\hat{\tau}(X)Y^0] + u_1 \mathbb{E}^{P_T}[-\hat{\tau}(X)Y^1]\right) + 2\left(u_0 \mathcal{V}^1(\hat{\tau}) + u_1 \mathcal{V}^0(\hat{\tau})\right) + \zeta. \qquad (10)$$

## 4.2 Establishing Distributionally Robust Metric

As Corollary 4.3 provides the distributionally robust (worst-case) value of PEHE, it can naturally measure the robustness of the CATE estimator $\hat{\tau}$ against distribution shift between counterfactual distribution and factual distribution. In this section, we will provide two steps involved in using Corollary 4.3 to construct the DRM method for CATE estimator selection.

**Step 1: Establishing computational tractability of $\mathcal{V}^t(\hat{\tau})$.** The distributionally robust values $\mathcal{V}^0(\hat{\tau})$ and $\mathcal{V}^1(\hat{\tau})$ in equation (10) are initially defined as supremum problems over infinite support, presenting a substantial computational challenge. Theorem 4.4 reformulates the infeasible supremum problems into tractable minimum problems.

**Theorem 4.4.** *The distributionally robust values $\mathcal{V}^0(\hat{\tau})$ and $\mathcal{V}^1(\hat{\tau})$ in equation (9) are equivalent to*

$$\mathcal{V}^0(\hat{\tau}) = \min_{\lambda_0 > 0} \lambda_0 \epsilon_0 + \lambda_0 \log \mathbb{E}^{P_C}[\exp(\hat{\tau}(X)Y^0/\lambda_0)];$$
$$\mathcal{V}^1(\hat{\tau}) = \min_{\lambda_1 > 0} \lambda_1 \epsilon_1 + \lambda_1 \log \mathbb{E}^{P_T}[\exp(-\hat{\tau}(X)Y^1/\lambda_1)]. \qquad (11)$$

*The proof is deferred to Appendix B.3.*

In the finite-sample scenario, $\mathcal{V}^0(\hat{\tau})$ and $\mathcal{V}^1(\hat{\tau})$ can be empirically approximated as follows:

$$\hat{\mathcal{V}}^0(\hat{\tau}) = \min_{\lambda_0 > 0} \lambda_0 \epsilon_0 + \lambda_0 \log \frac{1}{n_c} \sum_{i=1}^n (1 - T_i) \exp(\hat{\tau}(X_i)Y_i/\lambda_0);$$
$$\hat{\mathcal{V}}^1(\hat{\tau}) = \min_{\lambda_1 > 0} \lambda_1 \epsilon_1 + \lambda_1 \log \frac{1}{n_t} \sum_{i=1}^n T_i \exp(-\hat{\tau}(X_i)Y_i/\lambda_1). \qquad (12)$$

**Algorithm 1** Using DRM for CATE Estimator Selection

---

**Input:** The candidate CATE estimators $\{\hat{\tau}_1, \ldots, \hat{\tau}_J\}$. The validation dataset with $n$ i.i.d. observational samples $\{(X_i, T_i, Y_i)\}_{i=1}^n$. The number of iterations $K$. The initialization $\lambda_0^{(0)}$ and $\lambda_1^{(0)}$. The ambiguity radius $\epsilon_0$ and $\epsilon_1$.

1: **for** $j = 1$ to $J$ **do**
2:     **for** $k = 0$ to $K - 1$ **do**
3:         Compute $\hat{F}_t(\lambda_t^{(k)}, \epsilon_t; \hat{\tau}_j)$ for $t \in \{0, 1\}$ by equation (14a).
4:         Compute $\partial \hat{F}_t(\lambda_t^{(k)}, \epsilon_t; \hat{\tau}_j)/\partial \lambda_t^{(k)}$ for $t \in \{0, 1\}$ by equation (14b).
5:         $\lambda_t^{(k+1)} \leftarrow \max\{\lambda_t^{(k)} - \hat{F}_t(\lambda_t^{(k)}, \epsilon_t; \hat{\tau}_j)/(\partial \hat{F}_t(\lambda_t^{(k)}, \epsilon_t; \hat{\tau}_j)/\partial \lambda_t^{(k)}), 0\}$ for $t \in \{0, 1\}$.
6:         Save $\hat{\mathcal{V}}^t(\hat{\tau}_j)[k] = \hat{F}_t(\lambda_t^{(k+1)}, \epsilon_t; \hat{\tau}_j)$ for $t \in \{0, 1\}$.
7:     Return $\hat{\mathcal{V}}^t(\hat{\tau}_j) = \arg\min_{k \in \{0, \ldots, K-1\}} \hat{\mathcal{V}}^t(\hat{\tau}_j)[k]$ for $t \in \{0, 1\}$.
8:     Use $\hat{\mathcal{V}}^0(\hat{\tau}_j)$ and $\hat{\mathcal{V}}^1(\hat{\tau}_j)$ to compute $\mathcal{R}^{DRM}(\hat{\tau}_j)$ by equation (15).
**Output:** $\hat{\tau}_{select} = \arg\min_{\hat{\tau} \in \{\hat{\tau}_1, \ldots, \hat{\tau}_J\}} \mathcal{R}^{DRM}(\hat{\tau})$.

---

Note that in equation (12), the potential outcomes $Y^0$ and $Y^1$ are replaced by the observed outcome $Y$ due to the fact that $(1 - T)Y^0 = (1 - T)Y$ and $TY^1 = TY$, which aligns with the Consistency assumption in Assumption 2.1. We then provide a finite-sample analysis of the gap between $\hat{\mathcal{V}}^t(\hat{\tau})$ and $\mathcal{V}^t(\hat{\tau})$ in the following Theorem 4.5, which suggests the gap decays at a rate of $n^{-1/2}$.

**Theorem 4.5.** *Let $u_t := P(T = t)$ for $t \in \{0, 1\}$. Assume $0 < \underline{\lambda} \leq \lambda_0, \lambda_1 \leq \bar{\lambda}$ and $\hat{\tau}(X)Y$ is bounded within the range of $\underline{M}$ to $\bar{M}$. Define $C_{exp} = \mathbf{1}_{\{\underline{M} \leq \bar{M} \leq 0\}} \exp\left(\bar{M}/\bar{\lambda} - \underline{M}/\underline{\lambda}\right) + \mathbf{1}_{\{\underline{M} \leq 0, \bar{M} \geq 0\}} \exp\left(\bar{M}/\underline{\lambda} - \underline{M}/\underline{\lambda}\right) + \mathbf{1}_{\{0 \leq \underline{M} \leq \bar{M}\}} \exp\left(\bar{M}/\underline{\lambda} - \underline{M}/\bar{\lambda}\right)$. For $n \geq 2/u^2 \log(2/\delta)$ and $t \in \{0, 1\}$, with probability $1 - \delta$, we have*

$$|\hat{\mathcal{V}}^t(\hat{\tau}) - \mathcal{V}^t(\hat{\tau})| \leq \mathcal{O}\left(\sqrt{\frac{8\bar{\lambda}^2 \log \frac{2}{\delta}}{nu_t^2} C_{exp}^2}\right) + \mathcal{O}\left(\sqrt{\frac{2\bar{\lambda}^2 \log(\frac{2}{\delta})}{nu_t^2}}\right). \quad (13)$$

*The proof is deferred to Appendix B.4.*

**Step 2: Finalizing Distributionally Robust Metric for CATE estimator selection.** We first define two functions that are useful in obtaining $\mathcal{V}^0(\hat{\tau})$ and $\mathcal{V}^1(\hat{\tau})$:

$$\hat{F}_0(\lambda_0, \epsilon_0; \hat{\tau}) = \lambda_0 \epsilon_0 + \lambda_0 \log \frac{1}{n_c} \sum_{i=1}^{n_c} e^{\frac{z_i}{\lambda_0}}, \; \hat{F}_1(\lambda_1, \epsilon_1; \hat{\tau}) = \lambda_1 \epsilon_1 + \lambda_1 \log \frac{1}{n_t} \sum_{i=1}^{n_t} e^{\frac{-z_i}{\lambda_1}}; \quad (14a)$$

$$\frac{\partial \hat{F}_0}{\partial \lambda_0} = \epsilon_0 + \log \sum_{i=1}^{n_c} \frac{e^{\frac{z_i}{\lambda_0}}}{n_c} - \frac{\sum_{i=1}^{n_c} Z_i e^{\frac{z_i}{\lambda_0}}}{\lambda_0 \sum_{i=1}^{n_c} e^{\frac{z_i}{\lambda_0}}}, \; \frac{\partial \hat{F}_1}{\partial \lambda_1} = \epsilon_1 + \log \sum_{i=1}^{n_t} \frac{e^{\frac{-z_i}{\lambda_1}}}{n_t} - \frac{\sum_{i=1}^{n_t} -Z_i e^{\frac{-z_i}{\lambda_1}}}{\lambda_1 \sum_{i=1}^{n_t} e^{\frac{-z_i}{\lambda_1}}}. \quad (14b)$$

Here, $Z$ denotes $\hat{\tau}(X)Y$ for notational simplicity. We then use the Newton-Raphson method to find the empirical solution for $\hat{\mathcal{V}}^t(\hat{\tau})$, exploiting the convexity of $\hat{F}_t(\lambda_t, \epsilon_t; \hat{\tau})$ w.r.t. $\lambda_t$. Based on the distributionally robust value of PEHE, i.e., $\mathcal{V}_{PEHE}(\hat{\tau})$ in equation (10), we finally obtain the selected estimator $\hat{\tau}_{select} = \arg\min_{\hat{\tau} \in \{\hat{\tau}_1, \ldots, \hat{\tau}_J\}} \mathcal{R}^{DRM}(\hat{\tau})$ such that

$$\mathcal{R}^{DRM}(\hat{\tau}) = \frac{1}{n} \sum_{i=1}^n \hat{\tau}(X_i)^2 + \frac{2}{n} \left(\sum_{i=1}^{n_c} \hat{\tau}(X_i)Y_i + \sum_{i=1}^{n_t} -\hat{\tau}(X_i)Y_i + n_c \hat{\mathcal{V}}^1(\hat{\tau}) + n_t \hat{\mathcal{V}}^0(\hat{\tau})\right). \quad (15)$$

Algorithm 1 provides complete procedure of using the DRM method for CATE estimator selection.

**Discussion on the ambiguity radius $\epsilon$.** The ambiguity radius $\epsilon$ plays a critical role in real-world applications [54, 52, 60]. However, determining an appropriate value for $\epsilon$ can be challenging as it requires striking a balance between ensuring the bound in equation (9) holds and maintaining its tightness. Specifically, if $\epsilon$ is set too small, it fails to guarantee that the counterfactual distribution is contained within the ambiguity set centered at factual distribution (the bound in Corollary 4.3 can hold). On the other hand, if $\epsilon$ is set too large, even though the ambiguity set can encompass more

distributions to ensure the counterfactual distribution is contained, the bound in Corollary 4.3 can be less tight. In general, selecting a proper ambiguity radius is an open problem in distributioanlly robust optimization (DRO) literature [34, 54, 46, 48, 72].

In this paper, we provide a guidance for determining the ambiguity radius for our DRM method. Based on the above discussion, an ideal radius should be $\epsilon_1^* = D_{KL}(P_C||P_T)$ and $\epsilon_0^* = D_{KL}(P_T||P_C)$, which ensures that the bound in Corollary 4.3 holds and is tight. However, as defined in equation (8), both $P_C$ and $P_T$ involve counterfactual information, making it unattainable to directly compute $D_{KL}(P_C||P_T)$ and $D_{KL}(P_T||P_C)$. To overcome this challenge, we demonstrate that Proposition 4.6 provides an intriguing alternative approach to acquire $D_{KL}(P_C||P_T)$ and $D_{KL}(P_T||P_C)$ when unconfoundedness in Assumption 2.1 is satisfied.

**Proposition 4.6.** *Let $P_X^T := P(X|T = 1)$ and $P_X^C := P(X|T = 0)$ denote the covariates distribution in the treat and control group, respectively. Assuming that random variables $(X, T, Y^1, Y^0)$ satisfy the unconfoundedness in Assumption 2.1, we have*

$$D_{KL}(P_C||P_T) = D_{KL}(P_X^C||P_X^T); \quad D_{KL}(P_T||P_C) = D_{KL}(P_X^T||P_X^C). \tag{16}$$

*The proof is deferred to Appendix B.2.*

Proposition 4.6 provides an important insight that the uncomputable term $D_{KL}(P_C||P_T)$ (or $D_{KL}(P_T||P_C)$) can be replaced by a computable quantity $D_{KL}(P_X^C||P_X^T)$ (or $D_{KL}(P_X^T||P_X^C)$), where $P_X^C$ and $P_X^T$ are empirically observable. Consequently, the ideal ambiguity radius can be set as $\epsilon_1^* = D_{KL}(P_X^C||P_X^T)$ and $\epsilon_0^* = D_{KL}(P_X^T||P_X^C)$. While the KL divergence can be approximated using empirical algorithm (e.g, Nearest-Neighbor [73, 57]), we recommend setting the ambiguity radius larger than the empirically approximated KL divergence (see specific explanations in Appendix C.1). This is necessary because it ensures that the ambiguity set is large enough to contain the target distribution. It is also important to note that though the Algorithm 1 involves approximating $\epsilon_1^* = D_{KL}(P_X^C||P_X^T)$ and $\epsilon_0^* = D_{KL}(P_X^T||P_X^C)$, the DRM itself remains free of nuisances, as this approach only determines the ambiguity radius but does not involve learning any nuisance function such as the outcome function, propensity function, and plug-in learner.

## 5 Experiments

### 5.1 Experimental Setup.

**Estimators & Selectors.** We consider a total of **36 CATE estimators**, comprising the combination of 4 base ML models and 9 meta-learners. Specifically, the base ML models are Linear Regression (LR), Support Vector Machine (SVM), Random Forests (RF), and Neural Net (Net). We consider these ML models for CATE estimators because they are representative of both rigid and flexible models, with each encoded distinct inductive biases, as highlighted by [19, 20]. Note that for the LR method, we employ Ridge regression for regression tasks and Logistic regression for classification tasks. As for the remaining methods, we utilize their corresponding regressors and classifiers for regression and classification tasks, respectively. Regarding the meta-learners, we select a set of both traditional basic learners (S-, T-, PS-, and IPW-learners) and recently developed learners (X-, DR-, U-, R-, and RA-learners), as detailed in Appendix A.1. We consider **14 CATE selectors**, consisting of 9 plug-in methods that rely on the above 9 learners, 3 pseudo-outcome methods (pseudo-DR, -R, and -IF), the random selection, the factual selection (from the 6-learner pool with S-, T-), the Nearest-Neighbor Matching [62], and our proposed DRM. The specific details of baseline selectors are stated in Appendix A.2. We employ the eXtreme Gradient Boosting (XGB) [12] as the underlying ML model for both plug-in and pseudo-outcome methods. We choose XGB because: i) it demonstrates superior performance in various scenarios, ensuring a good performance of baseline selectors; ii) the need to avoid potential congeniality bias that may arise from using the similar ML models employed in CATE estimators [20]; iii) aligning with [5] where XGB is used for their proposed pseudo-IF metric. The details of hyperparameters for nuisance models are stated in Section C.2 of Appendix.

**Dataset.** Since the ground truth of CATE is unavailable in real-world data, previous studies commonly utilize semi-synthetic datasets to compare model performance. In line with [19, 20], we collect the covariates with $n = 4802$ data points from ACIC2016 dataset [22]. Then, we generate treatment with $T_i|X_i \sim Bern(1/(1 + \exp(-\xi(\beta_T' X_i + 3))))$, where $Bern$ indicates the Bernoulli

Table 1: Comparison of Regret for different selectors across Settings A, B, and C (Note that B ($\xi = 1$) matches A ($\rho = 0.1$)). Reported values (mean $\pm$ standard deviation) are computed over 100 experiments. Bold denotes the best three results among all selectors. Smaller value is better.

| | A ($\rho = 0$) | A ($\rho = 0.1$) | A ($\rho = 0.3$) | B ($\xi = 0$) | B ($\xi = 2$) | C ($m = 0.1$) | C ($m = 0.5$) | C ($m = 0.9$) |
|---|---|---|---|---|---|---|---|---|
| Plug-U | 47.87±94.89 | 39.22±60.78 | 32.13±61.49 | 0.51±2.09 | 151.98±291.20 | 39.41±52.58 | 54.47±209.86 | 15.58±26.41 |
| Plug-S | 3.38±7.73 | 2.65±5.65 | 2.25±5.64 | **0.22**±1.08 | **5.91**±10.61 | 2.72±4.70 | 3.74±6.34 | 4.65±6.33 |
| Plug-PS | 3.08±7.27 | 2.65±5.65 | 2.24±5.64 | **0.22**±1.08 | **5.91**±10.61 | 2.72±4.70 | 3.55±6.11 | 4.65±6.33 |
| Plug-T | 59.12±21.87 | 56.38±23.02 | 55.35±21.36 | 10.48±10.72 | 64.96±18.03 | 59.87±18.73 | 43.28±23.64 | 36.28±19.33 |
| Plug-X | 8.10±10.57 | 7.67±12.26 | 5.77±11.26 | 4.76±10.81 | 11.55±13.94 | 7.78±15.02 | 10.76±14.62 | 11.80±10.82 |
| Plug-IPW | 33.37±28.34 | 35.78±27.50 | 35.26±27.24 | 4.43±7.40 | 58.64±23.85 | 38.43±31.16 | 24.98±22.79 | 20.54±19.67 |
| Plug-DR | 43.11±26.54 | 43.76±26.92 | 44.20±26.48 | 4.22±7.97 | 64.60±18.88 | 46.57±32.93 | 28.66±23.08 | 23.59±18.21 |
| Plug-R | **1.92**±4.91 | **2.62**±15.63 | **1.47**±3.60 | 0.43±2.09 | 9.78±31.43 | **1.91**±4.94 | **2.53**±6.51 | **2.12**±4.94 |
| Plug-RA | 56.60±24.06 | 57.69±19.98 | 54.60±22.86 | 6.60±9.16 | 64.50±17.72 | 55.87±19.63 | 40.48±24.55 | 33.34±19.04 |
| Pseudo-DR | 61.35±22.41 | 61.09±20.08 | 59.06±19.46 | 14.75±22.95 | 70.02±17.53 | 62.08±19.65 | 48.83±25.78 | 44.99±23.53 |
| Pseudo-R | 9.85±27.04 | 14.12±45.74 | 5.94±21.05 | 4.73±20.40 | 14.86±30.64 | 10.58±24.63 | 15.93±29.84 | 21.26±32.51 |
| Pseudo-IF | 64.54±15.18 | 62.49±16.61 | 62.69±16.13 | 26.73±23.60 | 65.74±16.68 | 60.06±21.16 | 54.96±20.63 | 38.60±22.21 |
| Random | 7214±22745 | 6511±21651 | 4196±17049 | 1135±5596 | 7549±22498 | 3768±16625 | 6214±19942 | 3445±14591 |
| Fact | 51.09±18.00 | 50.86±19.40 | 51.01±21.03 | 14.33±16.54 | 65.23±27.53 | 48.92±17.19 | 47.40±22.51 | 40.37±23.14 |
| Matching | 60.85±21.45 | 62.18±17.77 | 59.91±18.67 | 13.33±22.86 | 68.98±17.24 | 61.52±19.04 | 52.83±23.85 | 40.01±24.05 |
| DRM | **0.96**±3.67 | **0.84**±4.83 | **1.25**±5.97 | 0.38±1.39 | 15.51±112.62 | **1.56**±8.73 | **1.40**±8.67 | **1.26**±3.52 |

distribution. The potential outcomes are generated by a linear function with interaction terms:

$$Y_i = \sum_j^d \beta_j' X_{i;j} + \sum_{j=1}^d \sum_{k=j}^d \beta_{j,k}' X_{i;j} X_{i;k} + \sum_{j=1}^d \sum_{k=j}^d \sum_{l=k}^d \beta_{j,k,l}' X_{i;j} X_{i;k} X_{i;l} + T_i \sum_{j=1}^d \gamma_j X_{i;j} + \epsilon_i.$$

The coefficient values are set as follows: $\beta_T, \beta_j, \beta_{j,k}, \beta_{j,k,l} \sim Bern(0.2)$, $\gamma_j \sim Bern(\rho)$, and $\epsilon_i \sim \mathcal{N}(0, 0.1)$. The parameter $\xi$ in treatment assignment represents the level of selection bias, and the parameter $\rho$ in $\gamma_j$ represents the complexity of the CATE function. We adopt the above data generating process to randomly generate 100 datasets, each with a training/validation/testing ratio of 49%/21%/30%.

**Settings.** In this section, we mainly investigate whether the estimator selected by DRM can demonstrate robustness to the selection bias and unobserved confounders. In addition, as demonstrated in [19, 20], the complexity of CATE function also affects relative performance of estimators and selectors. Given these considerations, we design the following three settings to compare the CATE selectors. **Setting A:** With the unconfoundedness assumption, let $\rho$ vary in $\{0, 0.1, 0.3\}$ with fixing $\xi = 1$. **Setting B:** With the unconfoundedness assumption, let $\xi$ vary in $\{0, 1, 2\}$ with fixing $\rho = 0.1$. **Setting C**: Without unconfoundedness assumption, fix $\rho = 0.1$ and $\xi = 1$. Then randomly remove $\lfloor m \cdot d \rfloor$ covariates such that the dimension of observed covariates is $d - \lfloor m \cdot d \rfloor$, where $m$ denotes the ratio of missing covariates varying in $\{0.1, 0.5, 0.9\}$. All the experiments are run on Dell 3640 with Intel Xeon W-1290P 3.60GHz CPU.

**Comparison criteria.** The CATE estimator $\hat{\tau}$ is believed better if it achieves a smaller difference between $\mathcal{R}^{oracle}(\hat{\tau})$ and $\mathcal{R}^{oracle}(\hat{\tau}_{best})$, where $\hat{\tau}_{best}$ is the actual best estimator in equation (3). We therefore use the following Regret criteria to compare estimators chosen by different selectors:

$$\text{Regret} = \mathcal{R}^{oracle}(\hat{\tau}_{select}) - \mathcal{R}^{oracle}(\hat{\tau}_{best}).$$

To further assess the ranking ability of each selector, we calculate the Spearman rank correlation between the rank order determined by the oracle metric $\mathcal{R}^{oracle}(\hat{\tau})$ and the rank order determined by each selector. All the reported values (Mean $\pm$ Standard deviation) are computed over 100 runs.

## 5.2 Experimental Results

**Regret comparison.** The results presented in Table 1 demonstrate consistently good performance from the DRM selector across various settings. In setting A, the DRM selector outperforms other selectors as the CATE complexity ($\rho$) varies. Additionally, Plug-R, Plug-S, and Plug-PS also perform well in terms of the Regret criterion, which aligns with prior findings in [66] that the R-objective is excellent in many cases. Note that the strong performance of Plug-S and Plug-PS may be due to less pronounced heterogeneity in the CATE function compared to the outcome function in the data generating process. We also compare the PEHE performance (i.e., $\mathcal{R}^{oracle}(\hat{\tau}_{select})$) of different selectors in Table 3 of Section C.3. The results indicate that Plug-R, Plug-S, and Plug-PS tend to

Table 2: Comparison of rank correlation for different selectors across Settings A, B, and C (Note that B ($\xi = 1$) matches A ($\rho = 0.1$)). Bold denotes the best three results among all selectors. Reported values (mean $\pm$ standard deviation) are computed over 100 experiments. Larger is better.

| | A ($\rho = 0$) | A ($\rho = 0.1$) | A ($\rho = 0.3$) | B ($\xi = 0$) | B ($\xi = 2$) | C ($m = 0.1$) | C ($m = 0.5$) | C ($m = 0.9$) |
|---|---|---|---|---|---|---|---|---|
| Plug-U | 0.69±0.34 | 0.70±0.35 | 0.75±0.29 | **0.95**±0.04 | 0.53±0.30 | 0.68±0.33 | 0.73±0.34 | 0.83±0.24 |
| Plug-S | 0.95±0.06 | 0.95±0.06 | 0.95±0.05 | **0.95**±0.04 | **0.95**±0.05 | 0.95±0.03 | 0.95±0.05 | 0.91±0.07 |
| Plug-PS | 0.95±0.06 | 0.95±0.06 | 0.95±0.05 | **0.95**±0.04 | **0.95**±0.05 | 0.95±0.03 | 0.95±0.05 | 0.91±0.07 |
| Plug-T | 0.54±0.18 | 0.54±0.18 | 0.54±0.16 | 0.89±0.07 | 0.57±0.16 | 0.51±0.16 | 0.58±0.21 | 0.59±0.21 |
| Plug-X | 0.94±0.05 | 0.94±0.04 | 0.94±0.04 | 0.93±0.05 | 0.93±0.05 | 0.93±0.04 | 0.92±0.06 | 0.85±0.13 |
| Plug-IPW | 0.72±0.19 | 0.71±0.19 | 0.71±0.19 | 0.92±0.06 | 0.68±0.15 | 0.69±0.19 | 0.76±0.18 | 0.77±0.17 |
| Plug-DR | 0.65±0.19 | 0.63±0.20 | 0.63±0.18 | 0.93±0.06 | 0.59±0.16 | 0.61±0.18 | 0.71±0.21 | 0.73±0.18 |
| Plug-R | **0.96**±0.03 | **0.96**±0.03 | **0.96**±0.03 | 0.95±0.04 | 0.93±0.07 | **0.96**±0.03 | **0.96**±0.05 | **0.96**±0.04 |
| Plug-RA | 0.55±0.19 | 0.54±0.17 | 0.55±0.17 | 0.92±0.06 | 0.57±0.15 | 0.53±0.17 | 0.60±0.22 | 0.62±0.21 |
| Pseudo-DR | 0.54±0.18 | 0.53±0.18 | 0.53±0.16 | 0.87±0.10 | 0.55±0.15 | 0.50±0.17 | 0.54±0.24 | 0.58±0.23 |
| Pseudo-R | 0.86±0.11 | 0.87±0.09 | 0.88±0.08 | 0.93±0.06 | 0.83±0.13 | 0.85±0.13 | 0.85±0.12 | 0.80±0.16 |
| Pseudo-IF | 0.52±0.17 | 0.52±0.17 | 0.51±0.15 | 0.66±0.18 | 0.64±0.16 | 0.52±0.16 | 0.53±0.19 | 0.62±0.18 |
| Random | 0.26±0.13 | 0.26±0.13 | 0.27±0.13 | 0.47±0.11 | 0.23±0.13 | 0.28±0.10 | 0.28±0.11 | 0.24±0.14 |
| Fact | 0.35±0.08 | 0.36±0.08 | 0.35±0.09 | 0.48±0.08 | 0.31±0.10 | 0.35±0.07 | 0.33±0.09 | 0.29±0.11 |
| Matching | 0.53±0.17 | 0.51±0.18 | 0.52±0.16 | 0.89±0.08 | 0.58±0.15 | 0.51±0.16 | 0.55±0.21 | 0.60±0.21 |
| DRM | 0.81±0.08 | 0.80±0.08 | 0.80±0.08 | 0.85±0.06 | 0.77±0.15 | 0.79±0.09 | 0.81±0.10 | 0.80±0.08 |

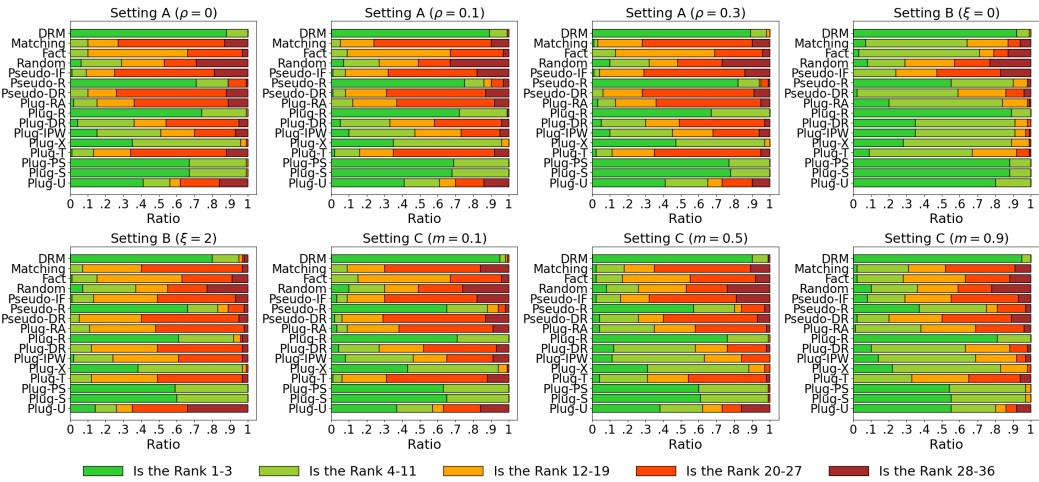

Figure 1: The stacked bar chart showing the distribution of the selected estimator's rank for each evaluation metric across rank intervals: [1-3], [4-11], [12-19], [20-27], and [28-36]. The greener (or redder) color indicates that the selected estimator ranks higher (or lower). For example, the **dark red** (or **green**) indicates the percentage of cases (out of 100 experiments) where the selected estimator ranks among the worst 9 estimators, specifically as ranks 28, 29, ..., or 36 (or among the best 3 estimators, specifically as ranks 1, 2, or 3).

exhibit better PEHE as the CATE complexity decreases, aligning with the findings in [20]. In setting B, the DRM selector demonstrates robustness against selection bias (controlled by $\xi$) compared to many baselines. However, for the case $\xi = 2$, DRM selects a poor estimator 1 or 2 times out of 100 experiments, as shown in Figure 1. Although this weakens its overall performance, DRM still outperforms many baselines in this scenario. In the scenario $\xi = 0$ where no selection bias is present, the factual selection criterion performs better in this specific setting. In this case, DRM does not demonstrate a significant advantage, as there is no distribution shift caused by selection bias. In setting C where the unconfoundedness assumption is violated, most selectors exhibit inferior performance. In contrast, DRM demonstrates consistent outperformance across all three cases, and its superiority becomes particularly significant as $m$ increases to 0.9, showcasing its robustness against the distribution shift arising from unobserved confounders.

**Ranking ability.** In Table 2, the DRM method demonstrates favorable performance in ranking estimators, surpassing certain Plug- (e.g., U, T, IPW, DR, RA) and Pseudo- (e.g., DR, IF) selectors.

In comparison to other nuisance-free baselines (Random, Fact, and Matching), DRM achieves significantly superior ranking ability. However, compared to Plug-S, -PS, -X, and -R, it does not exhibit remarkable performance in ranking CATE estimators, possibly due to the fact that DRM selects estimators based on their distributionally robust (worst-case) performance. Indeed, the definition of ranking inherently involves the concept of expected (average) performance, which is not determined solely by either the best or worst performance. While distributionally robust performance serves as a suitable criterion for selecting players to participate in the Olympics, it may not be a reasonable standard for ranking players' average performance. Therefore, it would be intriguing to explore some ways in future research that can enhance the ranking ability of our DRM selector.

**Variance analysis.** Table 1 indicates that baseline selectors tend to exhibit higher variances in Regret performance. This is primarily due to the wide range of PEHE performances across the 36 CATE estimators. If a selector consistently selects either good or bad estimators, the variance would not be very large. To investigate this further, we sorted all 36 estimators in ascending order based on their $\mathcal{R}^{oracle}(\hat{\tau})$ values, resulting in the sorted list: $[\mathcal{R}^{oracle}(\hat{\tau}_1), \ldots, \mathcal{R}^{oracle}(\hat{\tau}_J)]$. We then determine the actual rank of the selected estimator within this list and visualize the distribution of these 100 ranks using a stacked bar chart. Figure 1 shows that many baseline methods tend to select CATE estimators from various percentile ranges, leading to high variance across the 100 selections. Notably, the DRM selector consistently chooses higher-ranked (i.e., better performing in PEHE) estimators, demonstrating its robustness in CATE estimator selection.

**Potential improvements.** There are several potential improvements based on the current experimental settings. First, the existing results suggest that Plug-S performs better than Plug-T, indicating that the complexity of CATE function is relatively simple. It would help to provide more comprehensive analysis if investigating how DRM compares to baselines when the CATE function is more complex. Second, since the impact of selection bias can vary with sample size [3], it is important to compare different selectors when the sample size is sufficiently large. Third, considering baselines that are specifically designed for addressing hidden confounders could provide valuable insights for testing different selectors under such conditions. We encourage deeper investigation of causal model selection without assuming unconfoundedness. Finally, it would be good if future studies will apply DRM and other selectors in Healthcare, Economics, and Business applications with real-world data, as CATE estimator selection plays an important role in personalized decision makings.

## 6 Conclusion

This paper sheds lights on the potential of robustness in CATE estimator selection. We propose a distributionally robust metric (DRM). The proposed metric is nuisance-free, eliminating the need to fit models for nuisance parameters (outcome function, propensity function, and plug-in learner). Additionally, it is well-targeted for selecting a robust CATE estimator. We provide a finite sample analysis that demonstrates the gap between $\hat{\mathcal{V}}^t(\hat{\tau})$ and $\mathcal{V}^t(\hat{\tau})$ reduces at a rate of $n^{-1/2}$ for $t \in \{0, 1\}$. The experimental results showcase that the CATE estimator selected by DRM demonstrate robustness to the distribution shift incurred by covariate shift and hidden confounders.

**Limitations and future work.** This paper explores the potential of robustness in CATE estimator selection. However, we must acknowledge that our DRM method is not a silver bullet, as consistent estimation on the CATE are never attainable [14]. Here, we outline some challenges and suggest future research directions. First, while Proposition 4.6 provides useful guidance for setting ambiguity radius in the DRM algorithm, we cannot guarantee that the empirically-computed radius is optimal due to potential bias in the algorithm's approximation of KL-divergence. Second, as discussed in Section 5.2, enhancing the ranking capability of DRM is a promising area for further research. Moreover, our findings are based on KL-divergence. However, using other divergences, such as the Wasserstein distance, to construct the ambiguity set could incorporate more diverse distributions, despite the challenges in solving the dual formulation of the Wasserstein distributionally robust value. Simultaneously, exploring whether alternative divergences can yield a tighter bound for the PEHE error is also interesting [4]. Finally, inspired by [16], understanding how nuisance parameters influence metrics like plug-DR and pseudo-DR might be helpful in CATE estimator selection. We hope our methods and findings will spur interest in model selection for causal inference, as well as in related fields like domain adaptation and out-of-distribution generalization.

## Acknowledgement

Qi WU acknowledges the support from The CityU-JD Digits Joint Laboratory in Financial Technology and Engineering, The Hong Kong Research Grants Council [General Research Fund 11219420/9043008], and The CityU APRC Grant 9610643. The work described in this paper was partially supported by the InnoHK initiative, the Government of the HKSAR, and the Laboratory for AI-Powered Financial Technologies. We finally thank all the anonymous reviewers for their constructive suggestions.

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

# Appendix

## A CATE Estimation Strategies

### A.1 CATE Learners

We now detail how to construct CATE learners using the observed samples $\{(X_i, T_i, Y_i)\}_{i=1}^n$. Note that CATE learners are learned on the training set, so the sample size $n$ here equals the training sample size. Denote $n_t$ by the sample size in the treat group, and $n_c$ by the sample size in the control group such that $n = n_t + n_c$.

- S-learner: Let predictors=$(X, T)$, response=$Y$. Train a model $\hat{\mu}(X, T)$. Then we obtain $\hat{\tau}_S(X)$:

$$\hat{\tau}_S(X) = \hat{\mu}(X, 1) - \hat{\mu}(X, 0).$$

- T-learner: Let predictors=$X^T$ (covariates in the treat), response=$Y^T$ (outcome in the treat). Train a model $\hat{\mu}_1(X)$. Let predictors=$X^C$ (covariates in the control), response=$Y^C$ (outcome in the control). Train a model $\hat{\mu}_0(X)$. Then we obtain $\hat{\tau}_T(X)$:

$$\hat{\tau}_T(X) = \hat{\mu}_1(X) - \hat{\mu}_0(X).$$

- PS-learner: Fisrt-step: Train $\hat{\tau}_S(X)$ using the above-mentioned step in S-learner. Second-step: Let predictors=$X$, response=$\hat{\tau}_S(X)$. Train a model $\hat{\tau}_{PS}(X)$ from the following objective:

$$\hat{\tau}_{PS} = \arg\min_\tau \frac{1}{n} \sum_{i=1}^n (\tau(X_i) - \hat{\tau}_S(X_i))^2.$$

- IPW-learner: First-step: let predictors=$X$, response=$T$. Train a propensity score model $\hat{\pi}(X)$. Construct surrogate of CATE using pseudo-outcomes with inverse propensity weighting (IPW) formula: $Y_{IPW}^{1,0} = Y_{IPW}^1 - Y_{IPW}^0$, where $Y_{IPW}^1 = \frac{TY}{\hat{\pi}(X)}$ and $Y_{IPW}^0 = \frac{(1-T)Y}{1-\hat{\pi}(X)}$. Train a model $\hat{\tau}_{IPW}(X)$ from the following objective:

$$\hat{\tau}_{IPW} = \arg\min_\tau \frac{1}{n} \sum_{i=1}^n (\tau(X_i) - Y_{i,IPW}^{1,0})^2.$$

- X-learner [47]: First-step: Train $\hat{\mu}_1(X)$ and $\hat{\mu}_0(X)$ using the the above-mentioned procedure in T-learner. Train a propensity score model $\hat{\pi}(X)$ using the the above-mentioned procedure in IPW-learner. Second-step: Let predictors=$X^T$, response=$\hat{\mu}_1(X^T) - Y^T$, and predictors=$X^C$, response=$\hat{\mu}_0(X^C) - Y^C$. Obtain a model $\hat{\tau}_X(X)$ by learning two separate functions $\hat{\tau}_X^1(X)$ and $\hat{\tau}_X^0(X)$:

$$\hat{\tau}_X(X) = (1 - \hat{\pi}(X))\hat{\tau}_X^1(X) + \hat{\pi}(X)\hat{\tau}_X^0(X),$$

$$\hat{\tau}_X^1 = \arg\min_\tau \frac{1}{n_t} \sum_{i=1}^{n_t} (\tau(X_i) - (Y_i - \hat{\mu}_0(X_i)))^2,$$

$$\hat{\tau}_X^0 = \arg\min_\tau \frac{1}{n_c} \sum_{i=1}^{n_c} (\tau(X_i) - (\hat{\mu}_1(X_i) - Y_i))^2.$$

- U-learner [25, 55]: First-step: Let predictors=$X$, response=$Y$. Train a model $\hat{\mu}(X)$ to approximate the conditional mean outcome $\mathbb{E}[Y|X]$. Train a propensity score model $\hat{\pi}(X)$ using the the above-mentioned procedure in IPW-learner. Second-step: Compute the outcome residual $\xi = Y - \hat{\mu}(X)$ and treatment residual $\nu = T - \hat{\pi}(X)$. Train a model $\hat{\tau}_U(X)$ from the following objective:

$$\hat{\tau}_U = \arg\min_\tau \frac{1}{n} \sum_{i=1}^n (\frac{\xi_i}{\nu_i} - \tau(X_i))^2.$$

- DR-learner [41, 26]: First-step: Train $\hat{\mu}_1(X)$ and $\hat{\mu}_0(X)$ using the the above-mentioned procedure in T-learner. Train a propensity score model $\hat{\pi}(X)$ using the the above-mentioned procedure in IPW-learner. Second-step: Construct surrogate of CATE using pseudo-outcomes with doubly robust (DR) formula: $Y_{DR}^{1,0} = Y_{DR}^1 - Y_{DR}^0$, where $Y_{DR}^1 = \hat{\mu}_1(X) + \frac{T}{\hat{\pi}(X)}(Y - \hat{\mu}_1(X))$ and $Y_{DR}^0 = \hat{\mu}_0(X) + \frac{1-T}{1-\hat{\pi}(X)}(Y - \hat{\mu}_0(X))$. Train a model $\hat{\tau}_{DR}(X)$ from the following objective:

$$\hat{\tau}_{DR} = \arg\min_\tau \ \frac{1}{n}\sum_{i=1}^n (\tau(X_i) - Y_{i,DR}^{1,0})^2.$$

- R-learner [55]: First-step: Let predictors=$X$, response=$Y$. Train a model $\hat{\mu}(X)$ to approximate the conditional mean outcome $\mathbb{E}[Y|X]$. Train a propensity score model $\hat{\pi}(X)$ using the the above-mentioned procedure in IPW-learner. Second-step: Compute the outcome residual $\xi = Y - \hat{\mu}(X)$ and treatment residual $\nu = T - \hat{\pi}(X)$. Train a model $\hat{\tau}_R(X)$ from the following objective:

$$\hat{\tau}_R = \arg\min_\tau \ \frac{1}{n}\sum_{i=1}^n (\xi_i - \nu_i\tau(X_i))^2.$$

- RA-learner [18]: First-step: Train $\hat{\mu}_1(X)$ and $\hat{\mu}_0(X)$ using the the above-mentioned procedure in T-learner. Second-step: Construct surrogate of CATE using pseudo-outcomes with regression adjustment (RA) formula: $Y_{RA} = T(Y - \hat{\mu}_0(X)) + (1 - T)(\hat{\mu}_1(X) - Y)$. Train a model $\hat{\tau}_{RA}(X)$ from the following objective:

$$\hat{\tau}_{RA} = \arg\min_\tau \ \frac{1}{n}\sum_{i=1}^n (\tau(X_i) - Y_{i,RA})^2.$$

## A.2 CATE Selectors

We now detail how to construct CATE selectors using the observed samples $\{(X_i, T_i, Y_i)\}_{i=1}^n$. Note that CATE selectors are constructed on the validation set, so the sample size $n$ here equals the validation sample size.

- Plug-in selector: Obtain any CATE learners $\tilde{\tau}$ using the observational validation data. Then plug-in $\tilde{\tau}$ into the following metric $\mathcal{R}_{\tilde{\tau}}^{plug}(\hat{\tau})$:

$$\mathcal{R}_{\tilde{\tau}}^{plug}(\hat{\tau}) = \sqrt{\frac{1}{n}\sum_{i=1}^n (\hat{\tau}(X_i) - \tilde{\tau}(X_i))^2}.$$

For each plug-in selector $\tilde{\tau}$, the selected $j^*$-th CATE estimator is $\hat{\tau}_{j^*}$, where $j^* = \arg\min_{j \in \{1,...,J\}} \mathcal{R}_{\tilde{\tau}}^{plug}(\hat{\tau}_j)$.

- Pseudo-outcome selector:
  1. Pseudo-DR: Utilize validation data to estimate nuisance parameters $(\tilde{\mu}_1, \tilde{\mu}_0, \tilde{\pi})$, following the procedure described in Section A.1. $\tilde{Y}_{DR} = \tilde{Y}_{DR}^1 - \tilde{Y}_{DR}^0$, where $\tilde{Y}_{DR}^1 = \tilde{\mu}_1(X) + \frac{T}{\tilde{\pi}(X)}(Y - \tilde{\mu}_1(X))$ and $\tilde{Y}_{DR}^0 = \tilde{\mu}_0(X) + \frac{1-T}{1-\tilde{\pi}(X)}(Y - \tilde{\mu}_0(X))$. Then the pseudo-DR metric is

$$\mathcal{R}_{DR}^{pseudo}(\hat{\tau}) = \sqrt{\frac{1}{n}\sum_{i=1}^n (\hat{\tau}(X_i) - \tilde{Y}_{i,DR})^2}.$$

For pseudo-DR selector, the selected $j^*$-th CATE estimator is $\hat{\tau}_{j^*}$, where $j^* = \arg\min_{j \in \{1,...,J\}} \mathcal{R}_{DR}^{pseudo}(\hat{\tau}_j)$.

  2. Pseudo-R: Utilize validation data to estimate nuisance parameters $(\tilde{\mu}, \tilde{\pi})$, following the procedure described in Section A.1. Then the pseudo-R metric is

$$\mathcal{R}_R^{pseudo}(\hat{\tau}) = \sqrt{\frac{1}{n}\sum_{i=1}^n ((Y_i - \tilde{\mu}(X_i)) - \hat{\tau}(X_i)(T_i - \tilde{\pi}(X_i)))^2}.$$

For pseudo-R selector, the selected $j^*$-th CATE estimator is $\hat{\tau}_{j^*}$, where $j^* = \arg\min_{j\in\{1,...,J\}} \mathcal{R}_R^{pseudo}(\hat{\tau}_j)$.

3. Pseudo-IF [5]: Utilize validation data to estimate nuisance parameters $(\tilde{\mu}_1, \tilde{\mu}_0, \tilde{\pi})$, following the procedure described in Section A.1. Let $\tilde{\tau}(X) = (\tilde{\mu}_1(X) - \tilde{\mu}_0(X))$. Then the pseudo-IF metric is

$$\mathcal{R}_{IF}^{pseudo}(\hat{\tau}) = \sqrt{\frac{1}{n}\sum_{i=1}^{n}((1-B_i)\tilde{\tau}^2(X_i) + B_iY_i(\tilde{\tau}(X_i)-\hat{\tau}(X_i)) - A_i(\tilde{\tau}(X_i)-\hat{\tau}(X_i))^2 + \hat{\tau}^2(X_i)),}$$

where $A_i = T_i - \tilde{\pi}(X_i)$, $B_i = 2T_i(T_i - \tilde{\pi}(X_i))C_i^{-1}$, $C_i = \tilde{\pi}(X_i)(1 - \tilde{\pi}(X_i))$.

For pseudo-IF selector, the selected $j^*$-th CATE estimator is $\hat{\tau}_{j^*}$, where $j^* = \arg\min_{j\in\{1,...,J\}} \mathcal{R}_{IF}^{pseudo}(\hat{\tau}_j)$.

4. Other pseudo-outcome selector: By manipulating the formula of $\tilde{Y}$, it is possible to create additional pseudo-outcome selectors, such as the pseudo-IPW selector. In our paper, we choose pseudo-DR as the baseline because it is representative in the causal inference literature and it often demonstrates superior performance, owing to its doubly robust property.

# B  Proofs

## B.1  Proof of Proposition 4.1

*Proof.*

$\mathbb{E}[(\hat{\tau}(X) - \tau_{true}(X))^2]$

$= \mathbb{E}[(\hat{\tau}(X) - (\mu_1(X) - \mu_0(X)))^2]$

$= \mathbb{E}[(\hat{\tau}(X) - \mu_1(X) + \mu_0(X))^2]$

$= \mathbb{E}[(\hat{\tau}(X) - \mu_1(X))^2] + \mathbb{E}[\mu_0(X)^2] + 2\mathbb{E}[(\hat{\tau}(X) - \mu_1(X))\mu_0(X)]$

$= \mathbb{E}[\hat{\tau}(X)^2] + \mathbb{E}[\mu_1(X)^2] - 2\mathbb{E}[\hat{\tau}(X)\mu_1(X)] + \mathbb{E}[\mu_0(X)^2] + 2\mathbb{E}[\hat{\tau}(X)\mu_0(X)] - 2\mathbb{E}[\mu_1(X)\mu_0(X)]$

$= \mathbb{E}[\hat{\tau}(X)^2] - 2\mathbb{E}[\hat{\tau}(X)(\mu_1(X) - Y^1 + Y^1)] + 2\mathbb{E}[\hat{\tau}(X)(\mu_0(X) - Y^0 + Y^0)]$
$\quad + \mathbb{E}[\mu_1(X)^2] + \mathbb{E}[\mu_0(X)^2] - 2\mathbb{E}[\mu_1(X)\mu_0(X)]$

$= \mathbb{E}[\hat{\tau}(X)^2] - 2\mathbb{E}[\hat{\tau}(X)Y^1] - 2\mathbb{E}[\hat{\tau}(X)(\mu_1(X) - Y^1)] + 2\mathbb{E}[\hat{\tau}(X)Y^0] + 2\mathbb{E}[\hat{\tau}(X)(\mu_0(X) - Y^0)]$
$\quad + \mathbb{E}[\mu_1(X)^2] + \mathbb{E}[\mu_0(X)^2] - 2\mathbb{E}[\mu_1(X)\mu_0(X)]$

$= \mathbb{E}[\hat{\tau}(X)^2] - 2\mathbb{E}[\hat{\tau}(X)Y^1] - 2\mathbb{E}[\mathbb{E}[\hat{\tau}(X)\mu_1(X) - \hat{\tau}(X)Y^1|X]] + 2\mathbb{E}[\hat{\tau}(X)Y^0]$
$\quad + 2\mathbb{E}[\mathbb{E}[\hat{\tau}(X)\mu_0(X) - \hat{\tau}(X)Y^0|X]] + \mathbb{E}[\mu_1(X)^2] + \mathbb{E}[\mu_0(X)^2] - 2\mathbb{E}[\mu_1(X)\mu_0(X)]$

$= \mathbb{E}[\hat{\tau}(X)^2] - 2\mathbb{E}[\hat{\tau}(X)Y^1] - 2\mathbb{E}[\hat{\tau}(X)\mu_1(X) - \hat{\tau}(X)\mathbb{E}[Y^1|X]] + 2\mathbb{E}[\hat{\tau}(X)Y^0]$
$\quad + 2\mathbb{E}[\hat{\tau}(X)\mu_0(X) - \hat{\tau}(X)\mathbb{E}[Y^0|X]] + \mathbb{E}[\mu_1(X)^2] + \mathbb{E}[\mu_0(X)^2] - 2\mathbb{E}[\mu_1(X)\mu_0(X)]$

$= \mathbb{E}[\hat{\tau}(X)^2] - 2\mathbb{E}[\hat{\tau}(X)Y^1] - 2\mathbb{E}[\hat{\tau}(X)\mu_1(X) - \hat{\tau}(X)\mu_1(X)] + 2\mathbb{E}[\hat{\tau}(X)Y^0]$
$\quad + 2\mathbb{E}[\hat{\tau}(X)\mu_0(X) - \hat{\tau}(X)\mu_0(X)] + \mathbb{E}[\mu_1(X)^2] + \mathbb{E}[\mu_0(X)^2] - 2\mathbb{E}[\mu_1(X)\mu_0(X)]$

$= \mathbb{E}[\hat{\tau}(X)^2] + 2\mathbb{E}[\hat{\tau}(X)Y^0] - 2\mathbb{E}[\hat{\tau}(X)Y^1] + \mathbb{E}[\mu_1(X)^2] + \mathbb{E}[\mu_0(X)^2] - 2\mathbb{E}[\mu_1(X)\mu_0(X)]$

$= \mathbb{E}[\hat{\tau}(X)^2] + 2\mathbb{E}[\hat{\tau}(X)Y^0] - 2\mathbb{E}[\hat{\tau}(X)Y^1] + \zeta.$

$\qquad\qquad\qquad\qquad\qquad\qquad\qquad\qquad\qquad\qquad\qquad\qquad\qquad\qquad\qquad\qquad\square$

## B.2  Proof of Proposition 4.6

The following Proposition B.1 is useful in proving Proposition 4.6.

**Proposition B.1.** *Assuming the random variable tuple $(X, T, Y^1, Y^0)$ satisfies Assumption 2.1, we have*

$$p(X, Y^0, Y^1|T = 0) = p(Y^0, Y^1|X)p(X|T = 0);$$
$$p(X, Y^0, Y^1|T = 1) = p(Y^0, Y^1|X)p(X|T = 1). \tag{17}$$

*Proof.*

$$p(X, Y^0, Y^1 | T = 0)$$
$$= p(Y^0, Y^1 | X, T = 0) p(X | T = 0)$$
$$= p(Y^0, Y^1 | X) p(X | T = 0). \quad \text{(Unconfoundedness)}$$
$$p(X, Y^0, Y^1 | T = 1)$$
$$= p(Y^0, Y^1 | X, T = 1) p(X | T = 1)$$
$$= p(Y^0, Y^1 | X) p(X | T = 1). \quad \text{(Unconfoundedness)}$$

$\square$

Now we can prove Proposition 4.6.

*Proof.*
$$D_{KL}(P_C || P_T)$$
$$= D_{KL}(P(X, Y^0, Y^1 | T = 0) || P(X, Y^0, Y^1 | T = 1))$$
$$= \int_{\mathcal{X}} \int_{\mathcal{Y}^0} \int_{\mathcal{Y}^1} p(x, y^0, y^1 | T = 0) \log \frac{p(x, y^0, y^1 | T = 0)}{p(x, y^0, y^1 | T = 1)} dy^1 dy^0 dx$$
$$= \int_{\mathcal{X}} \int_{\mathcal{Y}^0} \int_{\mathcal{Y}^1} p(y^0, y^1 | x) p(x | T = 0) \log \frac{p(y^0, y^1 | x) p(x | T = 0)}{p(y^0, y^1 | x) p(x | T = 1)} dy^1 dy^0 dx \quad \text{(By Proposition B.1)}$$
$$= \int_{\mathcal{X}} \int_{\mathcal{Y}^0} \int_{\mathcal{Y}^1} p(y^0, y^1 | x) p(x | T = 0) \log \frac{p(x | T = 0)}{p(x | T = 1)} dy^1 dy^0 dx$$
$$= \int_{\mathcal{X}} \left( \int_{\mathcal{Y}^0} \int_{\mathcal{Y}^1} p(y^0, y^1 | x) dy^1 dy^0 \right) p(x | T = 0) \log \frac{p(x | T = 0)}{p(x | T = 1)} dx$$
$$= \int_{\mathcal{X}} p(x | T = 0) \log \frac{p(x | T = 0)}{p(x | T = 1)} dx$$
$$= D_{KL}(P(X | T = 0) || P(X | T = 1))$$
$$= D_{KL}(P_X^C || P_X^T).$$
Similarly, it is easy to show $D_{KL}(P_T || P_C) = D_{KL}(P_X^T || P_X^C)$

$\square$

## B.3 Proof of Theorem 4.4

**Lemma B.2** (Theorem 1 in [34]). *Let $f_\theta(X)$ denote the loss function of $X$ and it is bounded almost surely. $\theta \in \Theta$ represents the model parameters of the function $f_\theta(X)$. Let $\mathcal{B}_\epsilon(P)$ be the uncertainty ball centered at distribution $P$ with ambiguity radius $\epsilon$. Define $\kappa$ as the mass of the distribution $P$ on its essential supremum (Proposition 2 in [34]). Assume $f_\theta(X)$ is bounded and $\log \kappa + \epsilon < 0$, then we have*

$$\mathcal{V} := \sup_{Q \in \mathcal{B}_\epsilon(P)} \mathbb{E}^Q[f_\theta(X)] = \min_{\lambda > 0} \lambda \epsilon + \lambda \log \mathbb{E}^P[\exp(f_\theta(X)/\lambda)].$$

Our Theorem 4.4 follows by directly applying the above Lemma B.2.

## B.4 Proof of Theorem 4.5

For notational simplicity, we denote $W = (X, T, Y) \in \mathcal{W}$ and $Z = \hat{\tau}(X)Y$. Assume $Z$ is bounded within the range $\underline{M}$ and $\bar{M}$. Define the following functions:

$$G_0(\lambda_0; W) = \mathbb{E}[g_0(\lambda_0; W)], \quad \hat{G}_0(\lambda_0; W) = \frac{1}{n} \sum_{i=1}^n g_0(\lambda_0; W_i),$$
$$\text{where } g_0(\lambda_0; W) = (1 - T) \exp(Z/\lambda_0);$$

$$G_1(\lambda_1; W) = \mathbb{E}[g_1(\lambda_1; W)], \quad \hat{G}_1(\lambda_1; W) = \frac{1}{n} \sum_{i=1}^n g_1(\lambda_1; W_i),$$
$$\text{where } g_1(\lambda_1; W) = T \exp(-Z/\lambda_1).$$

Then we have the following lemma that guarantees the convergence for $\hat{G}_0(\lambda_0; W)$ and $\hat{G}_1(\lambda_1; W)$.

**Lemma B.3.** *Assume $0 < \underline{\lambda} \leq \lambda_0, \lambda_1 \leq \bar{\lambda}$, and $\hat{\tau}(X)Y$ is bounded within the range of $\underline{M}$ to $\bar{M}$. Then with probability $1 - \delta$, we have*

If $\underline{M} \leq \bar{M} \leq 0$ :

$$|\hat{G}_0(\lambda_0; W) - G_0(\lambda_0; W)| \leq \mathcal{O}\left(\sqrt{\frac{2\log\frac{2}{\delta}\left(\exp\left(\bar{M}/\bar{\lambda}\right)\right)^2}{n}}\right) ;$$

$$|\hat{G}_1(\lambda_1; W) - G_1(\lambda_1; W)| \leq \mathcal{O}\left(\sqrt{\frac{2\log\frac{2}{\delta}\left(\exp\left(-\underline{M}/\underline{\lambda}\right)\right)^2}{n}}\right) .$$

If $\underline{M} \leq 0, \bar{M} \geq 0$ :

$$|\hat{G}_0(\lambda_0; W) - G_0(\lambda_0; W)| \leq \mathcal{O}\left(\sqrt{\frac{2\log\frac{2}{\delta}\left(\exp\left(\bar{M}/\underline{\lambda}\right)\right)^2}{n}}\right) ;$$

$$|\hat{G}_1(\lambda_1; W) - G_1(\lambda_1; W)| \leq \mathcal{O}\left(\sqrt{\frac{2\log\frac{2}{\delta}\left(\exp\left(-\underline{M}/\underline{\lambda}\right)\right)^2}{n}}\right) . \tag{18}$$

If $0 \leq \underline{M} \leq \bar{M}$ :

$$|\hat{G}_0(\lambda_0; W) - G_0(\lambda_0; W)| \leq \mathcal{O}\left(\sqrt{\frac{2\log\frac{2}{\delta}\left(\exp\left(\bar{M}/\underline{\lambda}\right)\right)^2}{n}}\right) ;$$

$$|\hat{G}_1(\lambda_1; W) - G_1(\lambda_1; W)| \leq \mathcal{O}\left(\sqrt{\frac{2\log\frac{2}{\delta}\left(\exp\left(-\underline{M}/\bar{\lambda}\right)\right)^2}{n}}\right) .$$

*Proof.* Denote $h_0(W_1, W_2, \ldots, W_n) = \frac{1}{n}\sum_{i=1}^n g_0(\lambda_0; W_i)$. We notice that $h_0(W_1, W_2, \ldots, W_n)$ satisfies the bounded difference inequality:

$$\sup_{W_1,\ldots,W_n,W_i'\in\mathcal{W}} |h_0(W_1, \ldots, W_i, \cdots, W_n) - h_0(W_1, \ldots, W_i', \cdots, W_n)|$$

$$= \sup_{W_i,W_i'\in\mathcal{W}} \frac{|g_0(\lambda_0; W_i) - g_0(\lambda_0; W_i')|}{n}$$

$$\leq 2\sup_{W_i\in\mathcal{W}} \frac{|g_0(\lambda_0; W_i)|}{n} \leq \frac{2\exp\left(\bar{M}/\lambda_0\right)}{n}.$$

Note that $|\hat{G}_0(\lambda_0; W) - G_0(\lambda_0; W)| = |h_0(W_1, W_2, \ldots, W_n) - \mathbb{E}[h_0(W_1, W_2, \ldots, W_n)]|$. Then using McDiarmid's inequality, for any $\epsilon > 0$, we have

$$P\left(\left|\hat{G}_0(\lambda_0; W) - G_0(\lambda_0; W)\right| \geq \epsilon\right)$$

$$= P\left(|h_0(W_1, W_2, \ldots, W_n) - \mathbb{E}[h_0(W_1, W_2, \ldots, W_n)]| \geq \epsilon\right)$$

$$\leq 2\exp\left(-\frac{2\epsilon^2}{n(\frac{2\exp(\bar{M}/\lambda_0)}{n})^2}\right) = 2\exp\left(\frac{-n\epsilon^2}{2\left(\exp\left(\bar{M}/\lambda_0\right)\right)^2}\right) .$$

For some $\delta > 0$, we have

$$P\left(\left|\hat{G}_0(\lambda_0; W) - G_0(\lambda_0; W)\right| \geq \epsilon\right) \leq 2\exp\left(\frac{-n\epsilon^2}{2\left(\exp\left(\bar{M}/\lambda_0\right)\right)^2}\right) \leq \delta.$$

This solves $\epsilon$ such that

$$\epsilon \geq \sqrt{\frac{2\log\frac{2}{\delta}\left(\exp\left(\bar{M}/\lambda_0\right)\right)^2}{n}}.$$

The above inequality should hold for any $\lambda_0$ such that $0 < \underline{\lambda} \leq \lambda_0 \leq \bar{\lambda}$. Therefore, we have

$$\text{If } \bar{M} \geq 0: \quad \epsilon \geq \sqrt{\frac{2 \log \frac{2}{\delta} \left(\exp\left(\bar{M}/\underline{\lambda}\right)\right)^2}{n}};$$

$$\text{If } \bar{M} \leq 0: \quad \epsilon \geq \sqrt{\frac{2 \log \frac{2}{\delta} \left(\exp\left(\bar{M}/\bar{\lambda}\right)\right)^2}{n}}.$$

Similarly, denote $h_1(W_1, W_2, \ldots, W_n) = \frac{1}{n} \sum_{i=1}^n g_1(\lambda_1; W_i)$. We note that $h_1(W_1, W_2, \ldots, W_n)$ satisfies the bounded difference inequality:

$$\sup_{W_1,\ldots,W_n,W_i' \in \mathcal{W}} |h_1(W_1, \ldots, W_i, \cdots, W_n) - h_1(W_1, \ldots, W_i', \cdots, W_n)|$$

$$= \sup_{W_i, W_i' \in \mathcal{W}} \frac{|g_1(\lambda_1; W_i) - g_1(\lambda_1; W_i')|}{n}$$

$$\leq 2 \sup_{W_i \in \mathcal{W}} \frac{|g_1(\lambda_1; W_i)|}{n} \leq \frac{2 \exp\left(-\underline{M}/\lambda_1\right)}{n}.$$

Then using McDiarmid's inequality, for any $\epsilon > 0$, we have

$$P\left(\left|\hat{G}_1(\lambda_1; W) - G_1(\lambda_1; W)\right| \geq \epsilon\right)$$

$$= P\left(|h_1(W_1, W_2, \ldots, W_n) - \mathbb{E}[h_1(W_1, W_2, \ldots, W_n)]| \geq \epsilon\right)$$

$$\leq 2 \exp\left(-\frac{2\epsilon^2}{n(\frac{2\exp(-\underline{M}/\lambda_1)}{n})^2}\right) = 2 \exp\left(\frac{-n\epsilon^2}{2\left(\exp\left(-\underline{M}/\lambda_1\right)\right)^2}\right).$$

For some $\delta > 0$, we have

$$P\left(\left|\hat{G}_1(\lambda_1; W) - G_1(\lambda_1; W)\right| \geq \epsilon\right) \leq 2 \exp\left(\frac{-n\epsilon^2}{2\left(\exp\left(-\underline{M}/\lambda_1\right)\right)^2}\right) \leq \delta.$$

This solves $\epsilon$ such that

$$\epsilon \geq \sqrt{\frac{2 \log \frac{2}{\delta} \left(\exp\left(-\underline{M}/\lambda_1\right)\right)^2}{n}}.$$

The above inequality should hold for any $\lambda_1$ such that $0 < \underline{\lambda} \leq \lambda_1 \leq \bar{\lambda}$. Therefore, we have

$$\text{If } \underline{M} \geq 0: \quad \epsilon \geq \sqrt{\frac{2 \log \frac{2}{\delta} \left(\exp\left(-\underline{M}/\bar{\lambda}\right)\right)^2}{n}};$$

$$\text{If } \underline{M} \leq 0: \quad \epsilon \geq \sqrt{\frac{2 \log \frac{2}{\delta} \left(\exp\left(-\underline{M}/\underline{\lambda}\right)\right)^2}{n}}.$$

$\square$

In the following content, we will bound terms $\left|\log(\hat{G}_0(\lambda_0; W)) - \log\left(G_0(\lambda_0; W)\right)\right|$ and $\left|\log(\hat{G}_1(\lambda_1; W)) - \log\left(G_1(\lambda_1; W)\right)\right|$. Lemma B.4 is useful for bounding these two terms.

**Lemma B.4.** *Let $c$ be a constant. For any $x_1, x_2$ such that $x_1, x_2 \geq c > 0$, we have*

$$|\log(x_1) - \log(x_2)| \leq \frac{1}{c}|x_1 - x_2| \tag{19}$$

*Proof.* Without loss of generality, assume $0 < c \leq x_1 \leq x_2$. We then have

$$\log(x_2) - \log(x_1) = \log(\frac{x_2}{x_1}) = \log(1 + \frac{x_2}{x_1} - 1) \leq \frac{x_2}{x_1} - 1 = \frac{x_2 - x_1}{x_1} \leq \frac{x_2 - x_1}{c}.$$

Taking the absolute value of both the left-hand side and the right-hand side, we have

$$|\log(x_1) - \log(x_2)| \leq \frac{1}{c}|x_1 - x_2|.$$

$\square$

Next, we introduce Lemma B.5 that bounds terms $\left|\log(\hat{G}_0(\lambda_0; W)) - \log(G_0(\lambda_0; W))\right|$ and $\left|\log(\hat{G}_1(\lambda_1; W)) - \log(G_1(\lambda_1; W))\right|$.

**Lemma B.5.** *Let $u$ denote the probability of treat, i.e., $u = P(T = 1)$. Assume that $\lambda_0, \lambda_1 \in \Lambda := [\underline{\lambda}, \bar{\lambda}]$ and $\hat{\tau}(X)Y$ is bounded within $\underline{M}$ and $\bar{M}$. Then for $n \geq \max\{\frac{2}{u^2}\log\left(\frac{2}{\delta}\right), \frac{2}{(1-u)^2}\log\left(\frac{2}{\delta}\right)\}$, with probability $1 - \delta$, we have*

If $\underline{M} \leq \bar{M} \leq 0$ :

$$\left|\log(\hat{G}_0(\lambda_0; W)) - \log(G_0(\lambda_0; W))\right| \leq \frac{2}{\exp(\underline{M}/\underline{\lambda})(1-u)}\left|\hat{G}_0(\lambda_0; W) - G_0(\lambda_0; W)\right|;$$

$$\left|\log(\hat{G}_1(\lambda_1; W)) - \log(G_1(\lambda_1; W))\right| \leq \frac{2}{\exp(-\bar{M}/\bar{\lambda})u}\left|\hat{G}_1(\lambda_1; W) - G_1(\lambda_1; W)\right|.$$

If $\underline{M} \leq 0, \bar{M} \geq 0$ :

$$\left|\log(\hat{G}_0(\lambda_0; W)) - \log(G_0(\lambda_0; W))\right| \leq \frac{2}{\exp(\underline{M}/\underline{\lambda})(1-u)}\left|\hat{G}_0(\lambda_0; W) - G_0(\lambda_0; W)\right|; \quad (20)$$

$$\left|\log(\hat{G}_1(\lambda_1; W)) - \log(G_1(\lambda_1; W))\right| \leq \frac{2}{\exp(-\bar{M}/\underline{\lambda})u}\left|\hat{G}_1(\lambda_1; W) - G_1(\lambda_1; W)\right|.$$

If $0 \leq \underline{M} \leq \bar{M}$ :

$$\left|\log(\hat{G}_0(\lambda_0; W)) - \log(G_0(\lambda_0; W))\right| \leq \frac{2}{\exp(\underline{M}/\bar{\lambda})(1-u)}\left|\hat{G}_0(\lambda_0; W) - G_0(\lambda_0; W)\right|;$$

$$\left|\log(\hat{G}_1(\lambda_1; W)) - \log(G_1(\lambda_1; W))\right| \leq \frac{2}{\exp(-\bar{M}/\underline{\lambda})u}\left|\hat{G}_1(\lambda_1; W) - G_1(\lambda_1; W)\right|.$$

*Proof.* First, we bound the term $\left|\log(\hat{G}_0(\lambda_0; W)) - \log(G_0(\lambda_0; W))\right|$.

$G_0(\lambda_0; W)$ and $\hat{G}_0(\lambda_0; W)$ are greater than 0 and bounded because $Z = \hat{\tau}(X)Y$ is bounded within the range $\underline{M}$ and $\bar{M}$. Therefore, applying Lemma B.4, we have

$$\left|\log(\hat{G}_0(\lambda_0; W)) - \log(G_0(\lambda_0; W))\right| \leq \frac{1}{c}\left|\hat{G}_0(\lambda_0; W) - G_0(\lambda_0; W)\right|,$$

$$\text{where } c = \min\left\{\inf_{\lambda_0 \in \Lambda, W \in \mathcal{W}} \hat{G}_0(\lambda_0; W), \inf_{\lambda_0 \in \Lambda, W \in \mathcal{W}} G_0(\lambda_0; W)\right\}.$$

Moreover, for any $\lambda_0 \in \Lambda$, we have

If $\underline{M} \geq 0$ : $\quad G_0(\lambda_0; W) = \mathbb{E}[(1 - T)\exp(Z/\lambda_0)] = \mathbb{E}[\exp(Z/\lambda_0)|T = 0]P(T = 0)$
$$\geq \mathbb{E}[\exp(\underline{M}/\bar{\lambda})|T = 0](1 - u) = \exp(\underline{M}/\bar{\lambda})(1 - u);$$

$$\hat{G}_0(\lambda_0; W) = \frac{1}{n}\sum_{i=1}^{n}(1 - T_i)\exp(Z_i/\lambda_0) \quad (21)$$

$$\geq \frac{1}{n}\sum_{i=1}^{n}(1 - T_i)\exp(\underline{M}/\bar{\lambda}) = \exp(\underline{M}/\bar{\lambda})(1 - \hat{u}).$$

If $\underline{M} \leq 0$ : $\quad G_0(\lambda_0; W) = \mathbb{E}[(1 - T)\exp(Z/\lambda_0)] = \mathbb{E}[\exp(Z/\lambda_0)|T = 0]P(T = 0)$
$$\geq \mathbb{E}[\exp(\underline{M}/\underline{\lambda})|T = 0](1 - u) = \exp(\underline{M}/\underline{\lambda})(1 - u);$$

$$\hat{G}_0(\lambda_0; W) = \frac{1}{n}\sum_{i=1}^{n}(1 - T_i)\exp(Z_i/\lambda_0) \quad (22)$$

$$\geq \frac{1}{n}\sum_{i=1}^{n}(1 - T_i)\exp(\underline{M}/\underline{\lambda}) = \exp(\underline{M}/\underline{\lambda})(1 - \hat{u}).$$

Given $\hat{u} = \frac{1}{n}\sum_{i=1}^{n}T_i$ and $u = \mathbb{E}[\frac{1}{n}\sum_{i=1}^{n}T_i]$, using Hoeffding's inequality, we have

$$P\left(\left|\frac{1}{n}\sum_{i=1}^{n}(1 - T_i) - \mathbb{E}[\frac{1}{n}\sum_{i=1}^{n}(1 - T_i)]\right| \geq \frac{\mathbb{E}[\frac{1}{n}\sum_{i=1}^{n}(1 - T_i)]}{2}\right) \leq 2\exp\left(-\frac{2(\frac{1-u}{2})^2}{n(\frac{1}{n})^2}\right) \leq \delta.$$

We can solve $n$ by

$$2\exp\left(-\frac{n(1-u)^2}{2}\right) \leq \delta \Rightarrow n \geq \frac{2}{(1-u)^2}\log\left(\frac{2}{\delta}\right).$$

This indicates that $(1-\hat{u}) \geq (1-u)/2$ with probability $1-\delta$ when $n \geq \frac{2}{(1-u)^2}\log\left(\frac{2}{\delta}\right)$. Combining this with equations (21) and (22), with probability $1-\delta$, when $n \geq \frac{2}{(1-u)^2}\log\left(\frac{2}{\delta}\right)$, we have

If $\underline{M} \geq 0:$
$$\inf_{\lambda_0 \in \Lambda, W \in \mathcal{W}} G_0(\lambda_0; W) \geq \exp(\underline{M}/\bar{\lambda})(1-u);$$
$$\inf_{\lambda_0 \in \Lambda, W \in \mathcal{W}} \hat{G}_0(\lambda_0; W) \geq \exp(\underline{M}/\bar{\lambda})(1-\hat{u}) \geq \exp(\underline{M}/\bar{\lambda})(1-u)/2.$$

If $\underline{M} \leq 0:$
$$\inf_{\lambda_0 \in \Lambda, W \in \mathcal{W}} G_0(\lambda_0; W) \geq \exp(\underline{M}/\underline{\lambda})(1-u);$$
$$\inf_{\lambda_0 \in \Lambda, W \in \mathcal{W}} \hat{G}_0(\lambda_0; W) \geq \exp(\underline{M}/\underline{\lambda})(1-\hat{u}) \geq \exp(\underline{M}/\underline{\lambda})(1-u)/2.$$

Therefore, with probability $1-\delta$, when $n \geq \frac{2}{(1-u)^2}\log\left(\frac{2}{\delta}\right)$, we have

If $\underline{M} \geq 0:$
$$\left|\log(\hat{G}_0(\lambda_0; W)) - \log(G_0(\lambda_0; W))\right| \leq \frac{2}{\exp(\underline{M}/\bar{\lambda})(1-u)}\left|\hat{G}_0(\lambda_0; W) - G_0(\lambda_0; W)\right|;$$

If $\underline{M} \leq 0:$
$$\left|\log(\hat{G}_0(\lambda_0; W)) - \log(G_0(\lambda_0; W))\right| \leq \frac{2}{\exp(\underline{M}/\underline{\lambda})(1-u)}\left|\hat{G}_0(\lambda_0; W) - G_0(\lambda_0; W)\right|.$$

Next, we bound the term $\left|\log(\hat{G}_1(\lambda_1; W)) - \log(G_1(\lambda_1; W))\right|$. $G_1(\lambda_1; W)$ and $\hat{G}_1(\lambda_1; W)$ are greater than 0 and bounded above. Therefore, applying Lemma B.4, we have

$$\left|\log(\hat{G}_1(\lambda_1; W)) - \log(G_1(\lambda_1; W))\right| \leq \frac{1}{c}\left|\hat{G}_1(\lambda_1; W) - G_1(\lambda_1; W)\right|,$$
$$\text{where } c = \min\left\{\inf_{\lambda_1 \in \Lambda, W \in \mathcal{W}} \hat{G}_1(\lambda_1; W), \ \inf_{\lambda_1 \in \Lambda, W \in \mathcal{W}} G_1(\lambda_1; W)\right\}.$$

Moreover, for any $\lambda_1 \in \Lambda$, we have

If $\bar{M} \geq 0:$
$$G_1(\lambda_1; W) = \mathbb{E}[T\exp(-Z/\lambda_1)] = \mathbb{E}[\exp(-Z/\lambda_1)|T=1]P(T=1)$$
$$\geq \mathbb{E}[\exp(-\bar{M}/\underline{\lambda})|T=1]u = \exp(-\bar{M}/\underline{\lambda})u;$$
$$\hat{G}_1(\lambda_1; W) = \frac{1}{n}\sum_{i=1}^{n}T_i\exp(-Z_i/\lambda_1) \tag{23}$$
$$\geq \frac{1}{n}\sum_{i=1}^{n}T_i\exp(-\bar{M}/\underline{\lambda}) = \exp(-\bar{M}/\underline{\lambda})\hat{u}.$$

If $\bar{M} \leq 0:$
$$G_1(\lambda_1; W) = \mathbb{E}[T\exp(-Z/\lambda_1)] = \mathbb{E}[\exp(-Z/\lambda_1)|T=1]P(T=1)$$
$$\geq \mathbb{E}[\exp(-\bar{M}/\bar{\lambda})|T=1]u = \exp(-\bar{M}/\bar{\lambda})u;$$
$$\hat{G}_1(\lambda_1; W) = \frac{1}{n}\sum_{i=1}^{n}T_i\exp(-Z_i/\lambda_1) \tag{24}$$
$$\geq \frac{1}{n}\sum_{i=1}^{n}T_i\exp(-\bar{M}/\bar{\lambda}) = \exp(-\bar{M}/\bar{\lambda})\hat{u}.$$

Given $\hat{u} = \frac{1}{n}\sum_{i=1}^{n}T_i$ and $u = \mathbb{E}[\frac{1}{n}\sum_{i=1}^{n}T_i]$, using Hoeffding's inequality, we have

$$P\left(\left|\frac{1}{n}\sum_{i=1}^{n}T_i - \mathbb{E}[\frac{1}{n}\sum_{i=1}^{n}T_i]\right| \geq \frac{\mathbb{E}[\frac{1}{n}\sum_{i=1}^{n}T_i]}{2}\right) \leq 2\exp\left(-\frac{2(\frac{u}{2})^2}{n(\frac{1}{n})^2}\right) \leq \delta.$$

We can solve $n$ by

$$2\exp\left(-\frac{nu^2}{2}\right) \leq \delta \Rightarrow n \geq \frac{2}{u^2}\log\left(\frac{2}{\delta}\right).$$

This indicates that $hatu \geq u/2$ with probability $1-\delta$ when $n \geq \frac{2}{u^2}\log\left(\frac{2}{\delta}\right)$. Combining this with equations (23) and (24), with probability $1-\delta$, when $n \geq \frac{2}{u^2}\log\left(\frac{2}{\delta}\right)$, we have

If $\bar{M} \geq 0$: $\quad \inf_{\lambda_1 \in \Lambda, W \in \mathcal{W}} G_1(\lambda_1; W) \geq \exp(-\bar{M}/\underline{\lambda})u;$

$$\inf_{\lambda_1 \in \Lambda, W \in \mathcal{W}} \hat{G}_1(\lambda_1; W) \geq \exp(-\bar{M}/\underline{\lambda})\hat{u} \geq \exp(-\bar{M}/\underline{\lambda})u/2.$$

If $\bar{M} \leq 0$: $\quad \inf_{\lambda_1 \in \Lambda, W \in \mathcal{W}} G_1(\lambda_1; W) \geq \exp(-\bar{M}/\bar{\lambda})u;$

$$\inf_{\lambda_1 \in \Lambda, W \in \mathcal{W}} \hat{G}_1(\lambda_1; W) \geq \exp(-\bar{M}/\bar{\lambda})\hat{u} \geq \exp(\bar{M}/\bar{\lambda})u/2.$$

Therefore, with probability $1-\delta$, when $n \geq \frac{2}{u^2}\log\left(\frac{2}{\delta}\right)$, we have

If $\bar{M} \geq 0$:

$$\left|\log(\hat{G}_1(\lambda_1; W)) - \log\left(G_1(\lambda_1; W)\right)\right| \leq \frac{2}{\exp(-\bar{M}/\underline{\lambda})u}\left|\hat{G}_1(\lambda_1; W) - G_1(\lambda_1; W)\right|;$$

If $\bar{M} \leq 0$:

$$\left|\log(\hat{G}_1(\lambda_1; W)) - \log\left(G_1(\lambda_1; W)\right)\right| \leq \frac{2}{\exp(-\bar{M}/\bar{\lambda})u}\left|\hat{G}_1(\lambda_1; W) - G_1(\lambda_1; W)\right|.$$

This completes the proof of Lemma B.5. $\qquad\square$

Additionally, the following Lemma B.6 provides the bound of $|\log(\hat{u}) - \log(u)|$.

**Lemma B.6.** *Let* $\hat{u} = \frac{1}{n}\sum_{i=1}^{n} T_i$ *and* $u = \mathbb{E}[\frac{1}{n}\sum_{i=1}^{n} T_i]$. *For* $n \geq \frac{2}{u^2}\log\left(\frac{2}{\delta}\right)$, *with probability* $1-\delta$, *we have*

$$|\log(\hat{u}) - \log(u)| \leq \mathcal{O}\left(\sqrt{\frac{2\log(\frac{2}{\delta})}{nu^2}}\right). \tag{25}$$

*Proof.* Using Hoeffding's inequality, we have

$$P(|\hat{u} - u| \geq \epsilon) = P\left(\left|\frac{1}{n}\sum_{i=1}^{n} T_i - \mathbb{E}[\frac{1}{n}\sum_{i=1}^{n} T_i]\right| \geq \epsilon\right) \leq 2\exp\left(-2n\epsilon^2\right),$$

$$2\exp\left(-2n\epsilon^2\right) \leq \delta \quad \text{solves} \quad \epsilon \geq \sqrt{\frac{\log(\frac{2}{\delta})}{2n}}.$$

Notably, using the results in the previous lemma, we know for $n \geq \frac{2}{u^2}\log\left(\frac{2}{\delta}\right)$, $\hat{u} \geq u/2$. Therefore, we have

$$|\log(\hat{u}) - \log(u)| \leq \frac{1}{\min\{\hat{u}, u\}}|\hat{u} - u|. \quad \text{(By Lemma B.4)}$$

$$\leq \frac{2}{u}|\hat{u} - u| \leq \frac{2}{u}\mathcal{O}\left(\sqrt{\frac{\log(\frac{2}{\delta})}{2n}}\right) = \mathcal{O}\left(\sqrt{\frac{2\log(\frac{2}{\delta})}{nu^2}}\right).$$

$$\square$$

In the following, we will bound the term $|\hat{\mathcal{V}}(\hat{\tau}) - \mathcal{V}(\hat{\tau})|$ using above lemmas. We first define functions $F_0(\lambda_0)$, $\hat{F}_0(\lambda_0)$, $F_1(\lambda_1)$, and $\hat{F}_1(\lambda_1)$:

$$
\begin{aligned}
F_0(\lambda_0) &= \lambda_0 \epsilon_0 + \lambda_0 \log(\mathbb{E}^{P_C}[\exp(\hat{\tau}(X)Y/\lambda_0)]) \\
&= \lambda_0 \epsilon_0 + \lambda_0 \log\left(\frac{1}{1-u}\mathbb{E}[(1-T)\exp(\hat{\tau}(X)Y/\lambda_0)]\right); \\
\hat{F}_0(\lambda_0) &= \lambda_0 \epsilon_0 + \lambda_0 \log(\frac{1}{n_c}\sum_{i=1}^{n}(1-T_i)\exp(\hat{\tau}(X_i)Y_i/\lambda_0)) \\
&= \lambda_0 \epsilon_0 + \lambda_0 \log\left(\frac{1}{n(1-\hat{u})}\sum_{i=1}^{n}(1-T_i)\exp(\hat{\tau}(X_i)Y_i/\lambda_0)\right). \\
F_1(\lambda_1) &= \lambda_1 \epsilon_1 + \lambda_1 \log(\mathbb{E}^{P_T}[\exp(-\hat{\tau}(X)Y/\lambda_1)]) \\
&= \lambda_1 \epsilon_1 + \lambda_1 \log\left(\frac{1}{u}\mathbb{E}[T\exp(-\hat{\tau}(X)Y/\lambda_1)]\right); \\
\hat{F}_1(\lambda_1) &= \lambda_1 \epsilon_1 + \lambda_1 \log(\frac{1}{n_t}\sum_{i=1}^{n}T_i\exp(-\hat{\tau}(X_i)Y_i/\lambda_1)) \\
&= \lambda_1 \epsilon_1 + \lambda_1 \log\left(\frac{1}{n\hat{u}}\sum_{i=1}^{n}T_i\exp(-\hat{\tau}(X_i)Y_i/\lambda_1)\right).
\end{aligned}
$$

The following Lemma B.7 bounds the term $|\hat{F}(\lambda) - F(\lambda)|$.

**Lemma B.7.** *Let* $u := P(T = 1)$. *Assuming that* $0 < \underline{\lambda} \leq \lambda \leq \bar{\lambda}$ *and* $\hat{\tau}(X)Y$ *is bounded within the range of* $\underline{M}$ *to* $\bar{M}$. *Define* $C_{exp} = \mathbf{1}_{\{\underline{M} \leq \bar{M} \leq 0\}}\exp\left(\bar{M}/\bar{\lambda} - \underline{M}/\underline{\lambda}\right) + \mathbf{1}_{\{\underline{M} \leq 0, \bar{M} \geq 0\}}\exp\left(\bar{M}/\underline{\lambda} - \underline{M}/\underline{\lambda}\right) + \mathbf{1}_{\{0 \leq \underline{M} \leq \bar{M}\}}\exp\left(\bar{M}/\underline{\lambda} - \underline{M}/\bar{\lambda}\right)$. *For* $n \geq 2/u^2 \log(2/\delta)$, *with probability* $1 - \delta$, *we have*

$$
\begin{aligned}
|\hat{F}_0(\lambda_0) - F_0(\lambda_0)| &\leq \mathcal{O}\left(\sqrt{\frac{8\lambda_0^2 \log\frac{2}{\delta}}{n(1-u)^2}C_{exp}^2}\right) + \mathcal{O}\left(\sqrt{\frac{2\lambda_0^2 \log(\frac{2}{\delta})}{n(1-u)^2}}\right); \\
|\hat{F}_1(\lambda_1) - F_1(\lambda_1)| &\leq \mathcal{O}\left(\sqrt{\frac{8\lambda_1^2 \log\frac{2}{\delta}}{nu^2}C_{exp}^2}\right) + \mathcal{O}\left(\sqrt{\frac{2\lambda_1^2 \log(\frac{2}{\delta})}{nu^2}}\right).
\end{aligned}
\tag{26}
$$

*Proof.*

$$
\begin{aligned}
&|\hat{F}_0(\lambda_0) - F_0(\lambda_0)| \\
&= \left|\lambda_0\left(\log\left(\frac{1}{1-u}\mathbb{E}[(1-T)\exp(\hat{\tau}(X)Y/\lambda_0)]\right) - \log\left(\frac{1}{n(1-\hat{u})}\sum_{i=1}^{n}(1-T_i)\exp(\hat{\tau}(X_i)Y_i/\lambda_0)\right)\right)\right| \\
&= \lambda_0\left|\log\left(\mathbb{E}[(1-T)\exp(\hat{\tau}(X)Y/\lambda_0)]\right) - \log\left(\frac{1}{n}\sum_{i=1}^{n}(1-T_i)\exp(\hat{\tau}(X_i)Y_i/\lambda_0)\right) + \log(1-\hat{u}) - \log(1-u)\right| \\
&\leq \lambda_0\left|\log\left(\mathbb{E}[(1-T)\exp(\hat{\tau}(X)Y/\lambda_0)]\right) - \log\left(\frac{1}{n}\sum_{i=1}^{n}(1-T_i)\exp(\hat{\tau}(X_i)Y_i/\lambda_0)\right)\right| + \lambda_0\left|\log(1-\hat{u}) - \log(1-u)\right|.
\end{aligned}
$$

If $\underline{M} \le \bar{M} \le 0$ :

$|\hat{F}_0(\lambda_0) - F_0(\lambda_0)|$

$\le \dfrac{2\lambda_0}{\exp(\underline{M}/\underline{\lambda})(1-u)} \left| \hat{G}_0(\lambda_0; W) - G_0(\lambda_0; W) \right| + \lambda_0 \left| \log(1-\hat{u}) - \log(1-u) \right|$    (By Lemma B.5)

$\le \mathcal{O}\left( \sqrt{\dfrac{8\lambda_0^2 \log \frac{2}{\delta}}{n(1-u)^2} \left( \exp\left( \bar{M}/\bar{\lambda} - \underline{M}/\underline{\lambda} \right) \right)^2} \right) + \mathcal{O}\left( \sqrt{\dfrac{2\lambda_0^2 \log(\frac{2}{\delta})}{n(1-u)^2}} \right)$    (By Lemma B.3 and Lemma B.6)

If $\underline{M} \le 0, \bar{M} \ge 0$ :

$|\hat{F}_0(\lambda_0) - F_0(\lambda_0)|$

$\le \dfrac{2\lambda_0}{\exp(\underline{M}/\underline{\lambda})(1-u)} \left| \hat{G}_0(\lambda_0; W) - G_0(\lambda_0; W) \right| + \lambda_0 \left| \log(1-\hat{u}) - \log(1-u) \right|$    (By Lemma B.5)

$\le \mathcal{O}\left( \sqrt{\dfrac{8\lambda_0^2 \log \frac{2}{\delta}}{n(1-u)^2} \left( \exp\left( \bar{M}/\underline{\lambda} - \underline{M}/\underline{\lambda} \right) \right)^2} \right) + \mathcal{O}\left( \sqrt{\dfrac{2\lambda_0^2 \log(\frac{2}{\delta})}{n(1-u)^2}} \right)$    (By Lemma B.3 and Lemma B.6)

If $0 \le \underline{M} \le \bar{M}$ :

$|\hat{F}_0(\lambda_0) - F_0(\lambda_0)|$

$\le \dfrac{2\lambda_0}{\exp(\underline{M}/\bar{\lambda})(1-u)} \left| \hat{G}_0(\lambda_0; W) - G_0(\lambda_0; W) \right| + \lambda_0 \left| \log(1-\hat{u}) - \log(1-u) \right|$    (By Lemma B.5)

$\le \mathcal{O}\left( \sqrt{\dfrac{8\lambda_0^2 \log \frac{2}{\delta}}{n(1-u)^2} \left( \exp\left( \bar{M}/\underline{\lambda} - \underline{M}/\bar{\lambda} \right) \right)^2} \right) + \mathcal{O}\left( \sqrt{\dfrac{2\lambda_0^2 \log(\frac{2}{\delta})}{n(1-u)^2}} \right)$    (By Lemma B.3 and Lemma B.6)

$|\hat{F}_1(\lambda_1) - F_1(\lambda_1)|$

$= \left| \lambda_1 \left( \log\left( \dfrac{1}{u}\mathbb{E}[T \exp(-\hat{\tau}(X)Y/\lambda_1)] \right) - \log\left( \dfrac{1}{n\hat{u}} \sum_{i=1}^n T_i \exp(\hat{\tau}(X_i)Y_i/\lambda_0) \right) \right) \right|$

$= \lambda_1 \left| \log\left( \mathbb{E}[T \exp(\hat{\tau}(X)Y/\lambda_1)] \right) - \log\left( \dfrac{1}{n} \sum_{i=1}^n T_i \exp(\hat{\tau}(X_i)Y_i/\lambda_1) \right) + \log(\hat{u}) - \log(u) \right|$

$\le \lambda_1 \left| \log\left( \mathbb{E}[T \exp(\hat{\tau}(X)Y/\lambda_1)] \right) - \log\left( \dfrac{1}{n} \sum_{i=1}^n T_i \exp(\hat{\tau}(X_i)Y_i/\lambda_1) \right) \right| + \lambda_1 \left| \log(\hat{u}) - \log(u) \right|.$

If $\underline{M} \leq \bar{M} \leq 0$ :

$|\hat{F}_1(\lambda_1) - F_1(\lambda_1)|$

$$\leq \frac{2\lambda_1}{\exp(-\bar{M}/\bar{\lambda})u} \left| \hat{G}_0(\lambda_0; W) - G_0(\lambda_0; W) \right| + \lambda_1 \left| \log(\hat{u}) - \log(u) \right| \quad \text{(By Lemma B.5)}$$

$$\leq \mathcal{O}\left( \sqrt{\frac{8\lambda_1^2 \log \frac{2}{\delta}}{nu^2} \left( \exp\left( \bar{M}/\bar{\lambda} - \underline{M}/\underline{\lambda} \right) \right)^2} \right) + \mathcal{O}\left( \sqrt{\frac{2\lambda_1^2 \log(\frac{2}{\delta})}{nu^2}} \right) \quad \text{(By Lemma B.3 and Lemma B.6)}$$

If $\underline{M} \leq 0, \bar{M} \geq 0$ :

$|\hat{F}_1(\lambda_1) - F_1(\lambda_1)|$

$$\leq \frac{2\lambda_1}{\exp(-\bar{M}/\underline{\lambda})u} \left| \hat{G}_1(\lambda_1; W) - G_1(\lambda_1; W) \right| + \lambda_1 \left| \log(\hat{u}) - \log(u) \right| \quad \text{(By Lemma B.5)}$$

$$\leq \mathcal{O}\left( \sqrt{\frac{8\lambda_1^2 \log \frac{2}{\delta}}{nu^2} \left( \exp\left( \bar{M}/\underline{\lambda} - \underline{M}/\underline{\lambda} \right) \right)^2} \right) + \mathcal{O}\left( \sqrt{\frac{2\lambda_1^2 \log(\frac{2}{\delta})}{nu^2}} \right) \quad \text{(By Lemma B.3 and Lemma B.6)}$$

If $0 \leq \underline{M} \leq \bar{M}$ :

$|\hat{F}_1(\lambda_1) - F_1(\lambda_1)|$

$$\leq \frac{2\lambda_1}{\exp(-\bar{M}/\underline{\lambda})u} \left| \hat{G}_1(\lambda_1; W) - G_1(\lambda_1; W) \right| + \lambda_1 \left| \log(\hat{u}) - \log(u) \right| \quad \text{(By Lemma B.5)}$$

$$\leq \mathcal{O}\left( \sqrt{\frac{8\lambda_1^2 \log \frac{2}{\delta}}{nu^2} \left( \exp\left( \bar{M}/\underline{\lambda} - \underline{M}/\bar{\lambda} \right) \right)^2} \right) + \mathcal{O}\left( \sqrt{\frac{2\lambda_1^2 \log(\frac{2}{\delta})}{nu^2}} \right) \quad \text{(By Lemma B.3 and Lemma B.6)}$$

$\square$

Now, we can prove the result in Theorem 4.5.

*Proof.* Let $\hat{\lambda}_0 = \arg\min_\lambda \hat{F}_0(\lambda_0)$, $\lambda_0^* = \arg\min_{\lambda_0} F_0(\lambda_0)$, $\hat{\lambda}_1 = \arg\min_\lambda \hat{F}_1(\lambda_1)$ and $\lambda_1^* = \arg\min_{\lambda_1} F_1(\lambda_1)$. Then we have

$$\begin{aligned}
\mathcal{V}^0(\hat{\tau}) - \hat{\mathcal{V}}^0(\hat{\tau}) &= F_0(\lambda_0^*) - \hat{F}_0(\hat{\lambda}_0) \\
&= F_0(\lambda_0^*) - \hat{F}_0(\hat{\lambda}_0) + F_0(\hat{\lambda}_0) - F_0(\hat{\lambda}_0) \\
&= F_0(\hat{\lambda}_0) - \hat{F}_0(\hat{\lambda}_0) + F_0(\lambda_0^*) - F_0(\hat{\lambda}_0) \\
&\leq |F_0(\hat{\lambda}_0) - \hat{F}_0(\hat{\lambda}_0)| + 0 \\
&\leq \sup_{\lambda_0} |F_0(\lambda_0) - \hat{F}_0(\lambda_0)|.
\end{aligned}$$

$$\begin{aligned}
\hat{\mathcal{V}}^0(\hat{\tau}) - \mathcal{V}^0(\hat{\tau}) &= \hat{F}_0(\hat{\lambda}_0) - F_0(\lambda_0^*) \\
&= \hat{F}_0(\hat{\lambda}_0) - F_0(\lambda_0^*) + \hat{F}_0(\lambda_0^*) - \hat{F}_0(\lambda_0^*) \\
&= \hat{F}_0(\lambda_0^*) - F_0(\lambda_0^*) + \hat{F}_0(\hat{\lambda}_0) - \hat{F}_0(\lambda_0^*) \\
&\leq |\hat{F}_0(\lambda_0^*) - F_0(\lambda_0^*)| + 0 \\
&\leq \sup_{\lambda_0} |\hat{F}_0(\lambda_0) - F_0(\lambda_0)|.
\end{aligned}$$

$$\begin{aligned}
\mathcal{V}^1(\hat{\tau}) - \hat{\mathcal{V}}^1(\hat{\tau}) &= F_1(\lambda_1^*) - \hat{F}_1(\hat{\lambda}_1) \\
&= F_1(\lambda_1^*) - \hat{F}_1(\hat{\lambda}_1) + F_1(\hat{\lambda}_1) - F_1(\hat{\lambda}_1) \\
&= F_1(\hat{\lambda}_1) - \hat{F}_1(\hat{\lambda}_1) + F_1(\lambda_1^*) - F_1(\hat{\lambda}_1) \\
&\leq |F_1(\hat{\lambda}_1) - \hat{F}_1(\hat{\lambda}_1)| + 0 \\
&\leq \sup_{\lambda_1} |F_1(\lambda_1) - \hat{F}_1(\lambda_1)|.
\end{aligned}$$

$$\hat{\mathcal{V}}^1(\hat{\tau}) - \mathcal{V}^1(\hat{\tau}) = \hat{F}_1(\hat{\lambda}_1) - F_1(\lambda_1^*)$$
$$= \hat{F}_1(\hat{\lambda}_1) - F_1(\lambda_1^*) + \hat{F}_1(\lambda_1^*) - \hat{F}_1(\lambda_1^*)$$
$$= \hat{F}_1(\lambda_1^*) - F_1(\lambda_1^*) + \hat{F}_1(\hat{\lambda}_1) - \hat{F}_1(\lambda_1^*)$$
$$\leq |\hat{F}_1(\lambda_1^*) - F_1(\lambda_1^*)| + 0$$
$$\leq \sup_{\lambda_1} |\hat{F}_1(\lambda_1) - F_1(\lambda_1)|.$$

Therefore, we have

If $\underline{M} \leq \bar{M} \leq 0$ :

$$|\hat{\mathcal{V}}^0(\hat{\tau}) - \mathcal{V}^0(\hat{\tau})| \leq \sup_{\lambda} |\hat{F}(\lambda) - F(\lambda)| \leq \mathcal{O}\left(\sqrt{\frac{8\bar{\lambda}^2 \log\frac{2}{\delta}}{n(1-u)^2} \left(\exp\left(\bar{M}/\bar{\lambda} - \underline{M}/\underline{\lambda}\right)\right)^2}\right) + \mathcal{O}\left(\sqrt{\frac{2\bar{\lambda}^2 \log(\frac{2}{\delta})}{n(1-u)^2}}\right);$$

$$|\hat{\mathcal{V}}^1(\hat{\tau}) - \mathcal{V}^1(\hat{\tau})| \leq \sup_{\lambda} |\hat{F}(\lambda) - F(\lambda)| \leq \mathcal{O}\left(\sqrt{\frac{8\bar{\lambda}^2 \log\frac{2}{\delta}}{nu^2} \left(\exp\left(\bar{M}/\bar{\lambda} - \underline{M}/\underline{\lambda}\right)\right)^2}\right) + \mathcal{O}\left(\sqrt{\frac{2\bar{\lambda}^2 \log(\frac{2}{\delta})}{nu^2}}\right).$$

If $\underline{M} \leq 0, \bar{M} \geq 0$ :

$$|\hat{\mathcal{V}}^0(\hat{\tau}) - \mathcal{V}^0(\hat{\tau})| \leq \sup_{\lambda} |\hat{F}(\lambda) - F(\lambda)| \leq \mathcal{O}\left(\sqrt{\frac{8\bar{\lambda}^2 \log\frac{2}{\delta}}{n(1-u)^2} \left(\exp\left(\bar{M}/\underline{\lambda} - \underline{M}/\underline{\lambda}\right)\right)^2}\right) + \mathcal{O}\left(\sqrt{\frac{2\bar{\lambda}^2 \log(\frac{2}{\delta})}{n(1-u)^2}}\right);$$

$$|\hat{\mathcal{V}}^1(\hat{\tau}) - \mathcal{V}^1(\hat{\tau})| \leq \sup_{\lambda} |\hat{F}(\lambda) - F(\lambda)| \leq \mathcal{O}\left(\sqrt{\frac{8\bar{\lambda}^2 \log\frac{2}{\delta}}{nu^2} \left(\exp\left(\bar{M}/\underline{\lambda} - \underline{M}/\underline{\lambda}\right)\right)^2}\right) + \mathcal{O}\left(\sqrt{\frac{2\bar{\lambda}^2 \log(\frac{2}{\delta})}{nu^2}}\right).$$

If $0 \leq \underline{M} \leq \bar{M}$ :

$$|\hat{\mathcal{V}}^0(\hat{\tau}) - \mathcal{V}^0(\hat{\tau})| \leq \sup_{\lambda} |\hat{F}(\lambda) - F(\lambda)| \leq \mathcal{O}\left(\sqrt{\frac{8\bar{\lambda}^2 \log\frac{2}{\delta}}{n(1-u)^2} \left(\exp\left(\bar{M}/\underline{\lambda} - \underline{M}/\bar{\lambda}\right)\right)^2}\right) + \mathcal{O}\left(\sqrt{\frac{2\bar{\lambda}^2 \log(\frac{2}{\delta})}{n(1-u)^2}}\right);$$

$$|\hat{\mathcal{V}}^1(\hat{\tau}) - \mathcal{V}^1(\hat{\tau})| \leq \sup_{\lambda} |\hat{F}(\lambda) - F(\lambda)| \leq \mathcal{O}\left(\sqrt{\frac{8\bar{\lambda}^2 \log\frac{2}{\delta}}{nu^2} \left(\exp\left(\bar{M}/\underline{\lambda} - \underline{M}/\bar{\lambda}\right)\right)^2}\right) + \mathcal{O}\left(\sqrt{\frac{2\bar{\lambda}^2 \log(\frac{2}{\delta})}{nu^2}}\right).$$

Finally, we have

$$|\hat{\mathcal{V}}_t(\hat{\tau}) - \mathcal{V}_t(\hat{\tau})| \leq \mathcal{O}\left(\sqrt{\frac{8\bar{\lambda}^2 \log\frac{2}{\delta}}{nu_t^2} C_{exp}^2}\right) + \mathcal{O}\left(\sqrt{\frac{2\bar{\lambda}^2 \log(\frac{2}{\delta})}{nu_t^2}}\right).$$

Note that $u_1 = P(T = 1)$ and $u_0 = P(T = 0)$. $C_{exp} = \mathbf{1}_{\{\underline{M} \leq \bar{M} \leq 0\}} \exp\left(\bar{M}/\bar{\lambda} - \underline{M}/\underline{\lambda}\right) + \mathbf{1}_{\{\underline{M} \leq 0, \bar{M} \geq 0\}} \exp\left(\bar{M}/\underline{\lambda} - \underline{M}/\underline{\lambda}\right) + \mathbf{1}_{\{0 \leq \underline{M} \leq \bar{M}\}} \exp\left(\bar{M}/\underline{\lambda} - \underline{M}/\bar{\lambda}\right).$

$\square$

# C  Additional Materials

## C.1  Additional Explanations

**Q1. Why DRM can select CATE estimators that are robust to the uncertainty in PEHE caused by selection bias and unobserved confounders?**  In Section 3.1 and Section 4.1, we have presented theoretical explanations for the reason why DRM can measure a CATE estimator's robustness against selection bias and unobserved confounding. Below we will explain it more specifically.

In causal inference, all the CATE estimators are constructed on the observational factual data. But how reliable the CATE estimator that learned on factual data is? This question can be never known unless we have the knowledge of the oracle PEHE. As shown in equation (6), we know that the PEHE is equal to two $\hat{\tau}$-dependent terms, $\mathbb{E}[\hat{\tau}(X)Y^t|T = t]$ and $\mathbb{E}[\hat{\tau}(X)Y^t|T = 1 - t]$.

Unfortunately, $\mathbb{E}[\hat{\tau}(X)Y^t|T = 1 - t]$ is uncomputable empirically because we can only observe the factual distribution $P^F = P(X, Y^t|T = t)$ but not the counterfactual distribution $P^{CF} = P(X, Y^t|T = 1 - t)$. The unobserved counterfactual distribution can be regarded as an uncertain distribution varying around the observed and certain factual distribution $P^F$. If we could assume a "God's perspective" and observe $P^{CF}$ directly, the counterfactual distribution will be certain - like a quantum world! Such an uncertainty in $P^{CF}$ results in the uncertainty in PEHE. Now we will analyze the source of such uncertainty by analyzing the relationship between the uncertain distribution $P^{CF}$ and the certain distribution $P^F$ based on equation (2):

$$P(X, Y^t|T = 1 - t) = P(X, Y^t|T = t)\frac{P(Y^t|T = 1 - t, X)}{P(Y^t|T = t, X)}\frac{P(X|T = 1 - t)}{P(X|T = t)}.$$

From above, we find the unobservable distribution $P(X, Y^t|T = 1 - t)$ is equal to the observable distribution $P(X, Y^t|T = t)$ multiplied with $\frac{P(Y^t|T=1-t,X)}{P(Y^t|T=t,X)}\frac{P(X|T=1-t)}{P(X|T=t)}$. In other words, $\frac{p(y^t|T=1-t,x)}{p(y^t|T=t,x)}\frac{p(x|T=1-t)}{p(x|T=t)}$ controls the discrepancy between $P^F$ and $P^{CF}$. Note that if there is no unmeasured confounders, then we have $p(y^t|T = 1 - t, x) = p(y^t|T = t, x)$; and if there is no selection bias (covariate shift), then we have $p(x|T = 1 - t) = p(x|T = t)$. Now we understand the root cause of the discrepancy between $P^F$ and $P^{CF}$ (or between $\mathbb{E}[\tau(X)Y^t|T = 1 - t]$ and $\mathbb{E}[\tau(X)Y^t|T = t]$) lies at the unobserved confounders and selection bias (covariate shift). In the DRM method, the uncertainty caused by potential unobserved confounders and selection bias in PEHE can be further measured as the distributionally robust values $\hat{\mathcal{V}}^1$ and $\hat{\mathcal{V}}^0$. Then the PEHE w.r.t. the CATE estimator $\hat{\tau}$ will be at most $\mathcal{R}^{DRM}(\hat{\tau})$, as shown in equation (15). An estimator $\hat{\tau}$ that attains smallest $\mathcal{R}^{DRM}(\hat{\tau})$ by definition reflects the distributional robustness against potential unobserved confounders and selection bias.

**Q2. How to set $\epsilon^*$ when there are unobserved confounders?** When unobserved confounders are present, Proposition 3.6 can also provide guidance for setting $\epsilon^*$. Taking $\epsilon_1^* = D_{KL}(P_C||P_T)$ as an example, we have

$$D_{KL}(P_C||P_T)$$
$$= \int_{\mathcal{X}} \int_{\mathcal{Y}^0} \int_{\mathcal{Y}^1} p(y^0, y^1|x, T = 0)p(x|T = 0) \log \frac{p(y^0, y^1|x, T = 0)p(x|T = 0)}{p(y^0, y^1|x, T = 1)p(x|T = 1)} dy^1 dy^0 dx$$
$$= \int_{\mathcal{X}} \left(\int_{\mathcal{Y}^0} \int_{\mathcal{Y}^1} p(y^0, y^1|x, T = 0)dy^1 dy^0\right) p(x|T = 0) \log \frac{p(x|T = 0)}{p(x|T = 1)} dx$$
$$+ \int_{\mathcal{X}} \int_{\mathcal{Y}^0} \int_{\mathcal{Y}^1} p(y^0, y^1|x, T = 0)p(x|T = 0) \log \frac{p(y^0, y^1|x, T = 0)}{p(y^0, y^1|x, T = 1)} dy^1 dy^0 dx$$
$$= D_{KL}(P(X|T = 0)||P(X|T = 1))$$
$$+ \int_{\mathcal{X}} \int_{\mathcal{Y}^0} \int_{\mathcal{Y}^1} p(y^0, y^1|x, T = 0)p(x|T = 0) \log \frac{p(y^0, y^1|x, T = 0)}{p(y^0, y^1|x, T = 1)} dy^1 dy^0 dx$$
$$> D_{KL}(P_X^C||P_X^T)$$

Therefore, when unobserved confounders present, we can set $\epsilon^*$ to a larger value than the one guided by Proposition 4.6. Simultaneously, as the empirical approximation of $\epsilon_1^* = D_{KL}(P_X^C||P_X^T)$ and $\epsilon_0^* = D_{KL}(P_X^T||P_X^C)$ can be biased, we also suggest set $\epsilon^*$ to a large value than the empirically-computed ones to ensure the ambiguity set is large enough to contain the target distribution. Therefore, we generally set $\epsilon_1^* = D_{KL}(P_X^C||P_X^T) + 5.2$ and $\epsilon_0^* = D_{KL}(P_X^T||P_X^C) + 5.2$ for all settings in our experiment. Theoretically, a larger $\epsilon^*$ should guarantee the DRM-selected estimator to be more robust, as it allows for a broader range of possible counterfactual distributions in the ambiguity set. However, setting $\epsilon^*$ too large can result in overly conservative estimator selection (similar to the well-known accuracy-robustness tradeoff). Therefore, how to determine a proper ambiguity radius still remains an open challenge in both our work and distributionally robust optimization literature.

## C.2 Hyperparameters

- For linear model, we use LogisticRegressionCV and RidgeCV (both are with 3-fold cross-validation) from sklearn package to tune hyperparameters: Logistic regression: $Cs \in \{0.01, 0.1, 1, 10\}$; Ridge Regression: $\alpha \in \{0.01, 0.1, 1, 10, 100\}$.

- For Neural Net, we set the hidden layers as [200, 200, 200, 100, 100], each with the ReLU activation function. The model is trained using the Adam optimizer with a learning rate of 0.001, a batch size of 64, and 300 epochs.
- For RF, XGBoost, and SVM model, we use AutoML [71, 53] (with 3-fold cross-validation) from flaml package to tune hyperparameters.

## C.3 The Complementary Results

First, we would like to emphasize that the experimental results in this final version of the paper differ from those in the original version. We made several revisions based on the feedback from anonymous reviewers: 1) We add Neural Net model to the base ML models and add the U-learner to the meta-learners, increasing the number of CATE estimators from 24 to 36; 2) We adopted AutoML for hyperparameter tuning when training SVM, RF, and XGBoost. All code is available at `https://github.com/yiyhuang3/CATE_estimator_selection`.

Below, we prsent the complementary PEHE results for 36 candidate CATE estimators, where the candidate pool contains 4 ML models (LR, SVM, RF, and Neural Net) × 9 learners (U-, S-, T-, PS-, IPW-, X-, DR-, R-, RA-).

Table 3: Comparison of PEHE for different selectors across Settings A, B, and C (Note that B ($\xi = 1$) matches A ($\rho = 0.1$)), with base model for CATE estimator being {LR, SVM, RF, Net}. Reported values (mean $\pm$ standard deviation) are computed over 100 experiments. Smaller is better.

|  | A ($\rho = 0$) | A ($\rho = 0.1$) | A ($\rho = 0.3$) | B ($\xi = 0$) | B ($\xi = 2$) | C ($m = 0.1$) | C ($m = 0.5$) | C ($m = 0.9$) |
|---|---|---|---|---|---|---|---|---|
| Plug-U | 49.59±95.07 | 41.93±61.07 | 36.16±61.77 | 2.28±2.32 | 155.24±291.78 | 42.45±52.65 | 59.51±210.54 | 24.37±26.51 |
| Plug-S | 5.10±8.29 | 5.36±5.84 | 6.29±5.76 | **1.99**±1.41 | 9.18±11.57 | 5.76±5.46 | 8.78±7.55 | 13.45±9.53 |
| Plug-PS | 4.80±7.74 | 5.36±5.85 | 6.28±5.75 | **1.99**±1.41 | **9.17**±11.58 | 5.76±5.46 | 8.58±7.40 | 13.45±9.53 |
| Plug-T | 60.84±22.03 | 59.09±22.88 | 59.39±21.34 | 12.25±10.80 | 68.22±18.14 | 62.90±19.13 | 48.32±23.60 | 45.07±20.63 |
| Plug-X | 9.82±10.67 | 10.39±12.20 | 9.81±11.30 | 6.52±10.90 | 14.82±14.23 | 10.82±15.24 | 15.80±15.03 | 20.59±13.03 |
| Plug-IPW | 35.09±28.69 | 38.50±27.78 | 39.29±27.48 | 6.19±7.39 | 61.90±24.17 | 41.47±31.37 | 30.02±22.57 | 29.33±20.45 |
| Plug-DR | 44.83±26.77 | 46.47±27.02 | 48.23±26.56 | 5.98±8.09 | 67.87±18.94 | 49.61±33.34 | 33.69±23.05 | 32.39±19.28 |
| Plug-R | 3.64±5.01 | 5.33±15.72 | 5.51±3.75 | 2.19±2.31 | 13.04±31.51 | 4.95±5.38 | 7.56±7.88 | 10.91±7.92 |
| Plug-RA | 58.32±24.02 | 60.40±20.13 | 58.63±22.86 | 8.37±9.28 | 67.77±17.82 | 58.91±19.66 | 45.52±24.80 | 42.13±20.35 |
| Pseudo-DR | 63.07±22.54 | 63.80±20.22 | 63.10±19.41 | 16.51±23.05 | 73.29±17.48 | 65.12±20.14 | 53.87±26.16 | 53.79±24.91 |
| Pseudo-R | 11.57±27.25 | 16.83±45.81 | 9.97±21.23 | 6.49±20.46 | 18.13±30.61 | 13.62±24.78 | 20.96±30.42 | 30.05±32.02 |
| Pseudo-IF | 66.26±15.20 | 65.21±16.35 | 66.72±15.84 | 28.49±23.55 | 69.01±16.57 | 63.09±20.62 | 60.00±19.18 | 47.40±20.16 |
| Random | 7216±22745 | 6514±21650 | 4200±17048 | 1136±5595 | 7552±22498 | 3771±16625 | 6219±19942 | 3453±14590 |
| Fact | 52.81±18.01 | 53.58±19.42 | 55.05±21.10 | 16.09±16.50 | 68.50±27.69 | 51.96±17.45 | 52.44±22.51 | 49.16±24.47 |
| Matching | 62.57±21.57 | 64.90±17.85 | 63.94±18.72 | 15.10±22.93 | 72.25±17.46 | 64.56±19.59 | 57.87±24.40 | 48.81±25.23 |
| DRM | **2.68**±4.73 | **3.55**±5.65 | **5.28**±6.37 | 2.14±1.70 | 18.77±112.78 | **4.60**±9.58 | **6.44**±9.73 | **10.05**±7.19 |

