# OpenReview forum: "Unveiling the Potential of Robustness in Selecting Conditional Average Treatment Effect Estimators"
_NeurIPS.cc/2024/Conference — NeurIPS 2024 poster_

### Official Review · Reviewer_1tzr · 2024-07-10

**Soundness:** 4
**Presentation:** 2
**Contribution:** 4
**Rating:** 8
**Confidence:** 4

**Summary:**

This paper presents a method for selecting a conditional average treatment effect estimator using the distributional robust optimization (DRO) technique. The proposed method does not require specifying models for nuisance functions.

**Strengths:**

1. The result is strong, since this paper provides a method of evaluating multiple CATE estimators without specifying other baseline CATE estimator. As a result, more reliable assessment of CATE estimators is possible.
2. The empirical comparison is extensive. This paper evaluates many existing CATE estimator and transparently exhibits the performance of the proposed method.

Given that the paper contains several weaknesses discussed below, I think the paper’s result is interesting and powerful in practice.

**Weaknesses:**

**Misleading statements**

The following statement is generally not true.

> According to Rubin Causal Model [56], the CATE is determined by comparing potential outcomes under different treatment assignments (i.e., treat and control) for a specific individual.
>

I think the authors are confusing the “unit-level treatment effect” with the “conditional treatment effect”. The unit-level treatment effect $Y_i(T=1) - Y_i(T=0)$ “*for a specific individual*” $i$ cannot be estimated. The CATE is a group-level treatment effect, where the group is specified by the conditioned covariates. Please explicitly distinguish between the term “CATE” and an individual treatment effect.

**Noninformative Introduction**

The key research question in line 38, “Given multifarious options for CATE estimators, which should be chosen” are not well-defined in the Introduction section. As a reader, the minimum expectation for Introduction section is to see what were the limitations of previous methods with a simple example illustrating the limitations. However, by the design of the paper, until reaching to Section 3, Equation (2), it’s impossible to know (1) what are the plug-in and pseudo-outcome metrics and (2) what were their problems. The unclarity of the problem setting in Section 1 and 2, makes the first two sections non-informative.

**Redundancy**

I don’t see the point of categorizing previous methods as plug-in and pseudo-outcome. All previous methods rely on specifying nuisances, while the proposed method does not. I think such categorization is redundant and unnecessary for the literature review and explanation of the proposed method.

**Subjective statements**

I think this paper contains subjective evaluations of previous methods, which, as a reader, feels somewhat embarrassing. For example, in line 125, "previous high-quality paper" seems awkward and redundant — "previous paper" is enough. Similarly, "standing on the shoulders of giants" is also embarrassing. These kinds of subjective assessments of previous papers dilute the attractiveness of this paper and distract the authors from focusing on their own work.

**Questions:**

1. What is the relation with the paper [[1911.02029] Selective machine learning of doubly robust functionals (arxiv.org)](https://arxiv.org/abs/1911.02029)?
2. Is the empirical result matched with previous literature [58, 16, 45]?

**Limitations:**

The paper assumes the ignorability, which is pretty strong assumption in general.

---

> ### Author Rebuttal · Authors · 2024-08-02
>
> Dear Reviewer 1tzr,
>
> We sincerely appreciate your positive recognition of the soundness of our theoretical results and the extensiveness and transparency of our empirical experiments. We are also grateful for your valuable suggestions in improving our presentation and bringing new insights to future work. Below we will respond to your comments one by one.
>
> **W1. Distinguishing Conditional Average Treatment Effect (CATE) and Individualized Treatment Effect (ITE)**
>
> **R1:** We agree with your argument about the difference between CATE and ITE. In the revised version, we will clarify this in the Introduction, explicitly defining ITE as $Y^1-Y^0$ (unit-level) and CATE as $\mathbb{E}[Y^1-Y^0|X]$ (subgroup-level).
>
> **W2. The presentation of Introduction**
>
> **R2:** We will restructure it as follows:
> * Background on causal inference and CATE.
> * Motivation for CATE estimator selection and existing methods for CATE estimator selection.
> * Challenges faced by current methods (moving content from Section 3.1 here)
> * Introducing our DRM method.
>
> This restructuring will allow readers to quickly grasp existing CATE model selection metrics and their limitations before introducing our method, thus providing clearer motivation for our work. Thank you for your suggestion!
>
> **W3. Redundancy of categorizing previous methods as plug-in and pseudo-outcome**
>
> **R3:** Thank you for this comment, while we still think it is proper to keep such categorization for the following reasons.
>
> * This maintains consistency with [16, 45].
>
> * The plug-in and pseudo-outcome approaches differ fundamentally in their construction form, as detailed in Section A.2:
>
> 1. Plug-in metrics approximate the ground-truth CATE function using $\tilde{\tau}(X)$, which relies on off-the-shelf nuisance functions and only covariates variable $X$.
>
> 2. Pseudo-outcome metrics approximate the ground-truth CATE function using $\tilde{Y}(X,T,Y)$, incorporating off-the-shelf nuisance functions and variables $(X,T,Y)$.
>
> * Both DR-based and R-based objectives can be used to construct plug-in and pseudo-outcome metrics (see Section A.2). This categorization helps prevent potential confusion between Plug-R/Plug-DR and Pseudo-R/Pseudo-DR.
>
> We will emphasize their distinction in lines 115-123 of the revised paper.
>
> **W4. Subjective statements**
>
> **R4:** We appreciate this helpful suggestion for our presentation. We have removed the subjective statements and will ensure our tone remains objective and rigorous in future paper writing.
>
> **Q1. What is the relation with the paper [arxiv 1911.02029] Selective machine learning of doubly robust functionals?**
>
> **R1:**
>
> **(a) Difference**: While both papers address model selection in causal inference, they differ significantly in the interested quantity and selection objects:
>
> Interested quantity:
>
> * Our paper: Conditional Average Treatment Effect (CATE), i.e., $E[Y^1 - Y^0|X]$.
>
> * Their paper: Average Treatment Effect (ATE), i.e., $E[Y^1 - Y^0]$.
>
> Selection objects:
>
> * Our paper: Selects candidate CATE functions. The specific details are stated in Section 3, Background of CATE Estimator Selection.
>
> * Their paper: Selects candidate nuisance functions in the doubly robust (DR) estimator of ATE. The specific details are as follows:
>
> The DR estimator of ATE is
> $\theta_{DR}=\frac{1}{n}\sum_{i=1}^{n} [\mu(X_i,1)-\mu(X_i,0)+\frac{T_i}{\pi(X_i)}(Y-\mu(X_i,1))-\frac{1-T_i}{1-\pi(X_i)}(Y-\mu(X_i,0))]$.
> In $\theta_{DR}$, the nuisance function $\mu(X,T)$ can be fitted with different machine learners, e.g., Ridge regression, Support Vector Machine, Random Forest, etc, and the nuisance function $\pi(X)$ can be fitted with different machine learners, e.g., Logistic Regression, Support Vector Machine, Random Forest, etc. This learner-fitting necessity is the same as plug-in and pseudo-outcome methods in CATE estimator selection. But differently, the main goal of their paper is to select nuisance parameters $\mu$ and $\pi$ for the DR estimator of ATE, from $J_{\mu}$ candidate outcome nuisance functions $\mu_1(X,T), ..., \mu_{J_{\mu}}(X,T)$ and $J_{\pi}$ candidate propensity score nuisance functions $\pi_1(X), ..., \pi_{J_{\pi}}(X)$.
>
> **(b) Connection**:
> Interestingly, the nuisance function selection method in their paper offers insights for improving the baseline CATE selection methods Plug-DR and Pseudo-DR. Both Plug-DR and Pseudo-DR metrics involve the process of constructing $\tilde{Y}_i=\mu(X_i,1)-\mu(X_i,0)+\frac{T_i}{\pi(X_i)}(Y-\mu(X_i,1))-\frac{1-T_i}{1-\pi(X_i)}(Y-\mu(X_i,0))$ (see Sections A.1 and A.2). Note that similar to the mentioned paper, one also needs to select a machine learner to fit the nuisance functions $\mu(X,T)$ and $\pi(X)$ for constructing $\tilde{Y}$. Their approach could be helpful in choosing appropriate machine learning learners for fitting nuisance functions $\mu(X,T)$ and $\pi(X)$ in constructing $\tilde{Y}$. This opens an interesting avenue for future research, i.e., exploring nuisance function selection methods for Plug-DR and Pseudo-DR metrics in CATE model selection. We will discuss this interesting research direction in Related Work.
>
> **Q2.** Is the empirical result matched with [58, 16, 45]?
>
> **R2:** Our results align with some key findings from previous literature. For example, in lines 288-289, we observe the excellence of the R-objective in many scenarios, corroborating the findings in [58]. In lines 290-292, the results confirm the outperformance of R-based selectors as CATE complexity decreases, consistent with the conclusions in [16]. Notably, our analysis extends beyond previous work by examining CATE estimator performance against the level of unobserved confounders and selection bias, providing new insights into CATE estimator selection.
>
> Finally, we would like to thank you again for your time and effort in reviewing our paper. Your feedback is helpful and valuable in improving our work! We remain very open to discussing any additional comments you may have.

---

> > ### Comment · Reviewer_1tzr · 2024-08-10
> > **Response**
> >
> > Thank you for addressing my concerns and questions. I sincerely hope that my concerns will be fully addressed as promised, because I believe this paper is strong and deserves acceptance. Thank you for the additional experiments and detailed explanations. I will raise my score.

---

> ### Author Response · Authors · 2024-08-10
> **Thank you for your reply**
>
> Dear Reviewer 1tzr,
>
> Thank you so much for your recognition and appreciation! We will definitely address all your comments in the final version, as we also believe the suggestions are very important in improving our paper quality. Thank you again for your prompt reply and we truly appreciate your time and help during the whole review process!
>
> Best regards,
>
> Authors of paper 4584

---

### Official Review · Reviewer_9qbS · 2024-07-13

**Soundness:** 3
**Presentation:** 3
**Contribution:** 3
**Rating:** 8
**Confidence:** 4

**Summary:**

The paper addresses the challenging problem of Counterfactual Average Treatment Effect (CATE) model selection, where the goal is to develop a model selection metric using only observed data, as counterfactual labels are not available. Previous work has focused on model selection metrics that involve learning nuisance parameters on the validation set, making the task of choosing the appropriate CATE model selection criteria complex. The authors propose a solution with a model selection criterion, termed DRM, that avoids the need for training additional nuisance models and is designed to be robust to distribution shifts between treatment and control groups. They first derive the optimization objective for DRM and then propose a relaxation that is tractable with observed data samples. The paper includes a finite sample analysis of the proposed DRM estimator and benchmarks it against existing model selection criteria using synthetic datasets.

**Strengths:**

* The paper makes a significant contribution to CATE model selection by introducing the DRM criterion, which is notable for not requiring the training of additional nuisance models. Prior work often involves specific inductive biases (e.g., S-Learner, T-Learner, DRLearner) and necessitates training extra models, which can lead to sub-optimal choices and affect model selection performance. The DRM criterion addresses these limitations by not being tied to any particular CATE estimation design. Instead, it derives an upper bound on the ideal PEHE metric that is tractable with observed data alone, avoiding the need for additional nuisance models.

*  The theoretical formulation of DRM is compelling and enhances interpretability by linking the bound to the magnitude of the shift between control and treatment distributions. This directly addresses the challenge of distribution shift in effect estimation. Additionally, the authors derive an equivalent formulation in terms of the KL divergence between the control covariates ($P(X|T=0$) and treatment covariates ($P(X|T=1$), which can be computed using observed data. This approach provides a practical way to assess the impact of distribution shift on CATE model selection.

* The derivation of the DRM criteria from the PEHE metric, along with the proposed estimation strategy, appears to be novel to the best of my knowledge.

* The paper is well-written, with the methodology behind DRM presented in an accessible manner. The background section effectively motivates their approach, and the experimental results are clearly presented, accompanied by a thorough discussion of the performance compared to baselines.

**Weaknesses:**

While the authors have done a good job in the experiments by considering diverse synthetic datasets and interesting ablation studies, I think the empirical analysis can still be improved. I have summarized my concerns with the experiments below, which would help to provide a clearer understanding of the performance differences and strengthen the comparison of DRM with other model selection criteria.

* The authors use only three base ML models to train the CATE estimators, which limits the diversity of models considered for model selection. To make the model selection problem more challenging and provide a more thorough evaluation, it would be beneficial to train a larger number of CATE estimators. This would offer a more comprehensive assessment of the DRM criterion's performance and robustness across a broader range of models.

* Estimating the nuisance parameters for model selection metrics is crucial for their performance, as highlighted by Mahajan et al. [1], who showed that plug-in metrics based on T-Learner/X-Learner are effective for model selection. To ensure a fair analysis, the authors should consider using state-of-the-art techniques, such as AutoML, for estimating the nuisance parameters associated with the model selection metrics. Currently, the performance of many model selection metrics might be suboptimal due to potentially poor choices of nuisance parameters

* The comparison of different model selection criteria in the results raises some concerns, particularly due to the high variance observed in the performance of several baselines. This variance might stem from poor results on a few datasets, while these baselines could be comparable to DRM on a majority of datasets. The authors should clarify this in the paper and offer more detailed insights, such as results broken down by dataset or other methods to highlight trends and performance variations.

References:

[1] Mahajan, Divyat, Ioannis Mitliagkas, Brady Neal, and Vasilis Syrgkanis. "Empirical Analysis of Model Selection for Heterogeneous Causal Effect Estimation." arXiv preprint arXiv:2211.01939 (2022)

**Questions:**

* In Table 1, I don't understand why DRM is more robust to selection bias as stated by the authors. Are authors comparing the performance gains with DRM over the best baselines across scenarios ($\mathcal{E}= \{0, 1, 2\}$)? It doesn't seem like the performance gap becomes better with more selection bias. Also, the performance of Plug-R or other baselines across different scenarios ($\mathcal{E}= \{0, 1, 2\}$) is statistically indistinguishable given the difference in means lies within the confidence interval. So I am not sure whether any of the prior metrics are susceptible to selection bias in this empirical study.

* Why is the Plug-T metric much worse than the Plug-S metric? T-Learner should provide more flexibility in choosing the regression models than S-Learner, so I don't understand why would the estimates of the ground truth be worse with T-Learner (hence worse model selection).

**Limitations:**

Yes, the authors have properly addressed the limitations of their work.

---

> ### Author Rebuttal · Authors · 2024-08-05
>
> Dear Reviewer 9qbs,
>
> Thank you for your thorough review of our paper. We are delighted that you recognize the novelty and significance of our DRM method in advancing CATE model selection, particularly its nuisance-free and robustness to distribution shift. Below we will address your comments.
>
> * Suggestions:
>
> **S1.** Considering a larger set of base models for CATE estimators.
>
> **R1.** As you may note, prior research has conducted comprehensive empirical investigations conducted a thorough and comprehensive empirical investigation, providing valuable insights for baseline CATE selectors. In our work, our primary focus has been on proposing a new metric for CATE estimator selection, and we believe the 24 CATE estimators with 8 widely-used meta-learners and 3 base ML models were adequate for the current examination. However, we do agree that considering a larger set of base models could offer a more comprehensive analysis. As such, we have expanded our experiments to include an additional Neural Net model, and the results are reported in Table 4 of the rebuttal pdf, and the updated results will be added into the revised paper. Specifically, the Net has 5 hidden layers [200, 200, 200, 100, 100], each with the ReLU activation function. The model is trained using the Adam optimizer with a learning rate of 0.001, a batch size of 64, and 300 epochs.
>
> **S2.** Using AutoML for estimating nuisance parameters.
>
> **R2.** In the original paper, we have tuned hyperparameters for each model in line with [16]. The details are discussed in GR3 of the top-page rebuttal. Thank you for recommending AutoML. A recent ICLR paper [45] has shown the benefits of using AutoML to search a broader grid of hyperparameters, ensuring well-trained nuisance functions. We will use AutoML to update our results and also in future research for hyperparameter tuning.
>
> **S3.** The high variance phenomenon should be further discussed.
>
> **R3.** Thank you for this insightful suggestion. The high variance phenomenon is discussed in GR2 of the top-page rebuttal and the discussion will be added into the revised paper.
>
> * Questions
>
> **Q1.** Not sure whether any of the prior metrics are susceptible to selection bias. Why DRM is robust to selection bias compared to baselines?
>
> **R1.** It is widely acknowledged in the causal inference literature that increasing selection bias can lead to larger biases in estimators and selectors (e.g., [8, 30, 67, 68]). Accordingly, for most selectors, the regret is expected to increase with higher selection bias - a trend clearly evident in the results presented in Table 1. Therefore, we would not expect a regret-decreasing trend with higher selection bias for most selectors, including our DRM method.
>
> DRM metric is robust to selection bias when comparing its performance to other baseline selectors, such as Plug-R. Specifically, as the level of selection bias increases from $\xi=1$ to $\xi=2$, the average regret of Plug-R increases significantly, from 1.27 to 4.38 - a jump of 3.11. In contrast, the average regret of DRM only increases from 0.11 to 1.21, a much smaller change of 1.10. However, it is important to note that in the case of no selection bias ($\xi=0$), DRM does not show a clear advantage over some baselines. This aligns with our expectations because DRM is designed to select estimators that are robust to distribution shifts.
>
> **Q2.** Why is Plug-T worse than Plug-S?
>
> **R2.** The key factor affecting the performance of Plug-T relative to Plug-S appears to be the level of heterogeneity. Heterogeneity refers to the complexity of the CATE function relative to the potential outcome functions. In our paper, the parameter $\rho$ controls the degree of heterogeneity. According to Table 1, the average regret gap between Plug-T and Plug-S decreases as $\rho$ increases from 0 to 0.3. Specifically, the regret gap decreases from 38.12 - 3.40 = 34.72 to 34.38 - 2.34 = 32.04 as heterogeneity increases. This suggests a trend that Plug-T would perform on par with (or better than) Plug-S as the level of heterogeneity increases.
>
> This observation aligns with insights from prior literature. For example, the authors in [16] found that the T-learner outperforms the S-learner in terms of PEHE when the heterogeneity is much larger. Additionally, [40] noted that the S-learner performs better when the CATE complexity is lower than the potential outcome functions, while the T-learner is preferred when the CATE is more complex. Furthermore, the authors in [45] used the original ACIC data but discarded instances with low-variance CATE to ensure sufficient heterogeneity, where they found that T-learner outperforms S-learner.
>
> Therefore, based on our results and the insights from previous studies, we believe the level of heterogeneity is a key factor that affects the relative performance of Plug-T and Plug-S. It would be interesting to further investigate other possible factors that may influence their comparative performance, as they are widely used meta-learners in real-world applications.
>
> Finally, we would like to express our sincere gratitude for your time and effort in reviewing our paper and providing valuable suggestions and feedback! We are very open to addressing and discussing any further comments or questions you may have.

---

> > ### Comment · Reviewer_9qbS · 2024-08-10
> >
> > Thanks for the detailed response to my questions! My concerns have been addressed and I have updated my rating accordingly.

---

> ### Author Response · Authors · 2024-08-10
> **Thank you for your reply**
>
> Dear Reviewer 9qbS,
>
> Thank you so much for your recognition of our paper and rebuttal! We will carefully incorporate your suggestions in finalizing our paper. We would like to express our gratitude again for your time and effort in the review process!
>
> Best regards,
>
> Authors of paper 4584

---

### Official Review · Reviewer_e7g2 · 2024-07-15

**Soundness:** 3
**Presentation:** 3
**Contribution:** 2
**Rating:** 6
**Confidence:** 3

**Summary:**

The paper introduces a new metric for CATE model evaluation and selection. Specifically, they derive a distributionally robust metric (DRM) which is nuisance-free and robust against selection bias. They show and explain its robust performance in extensive benchmarking experiments against existing baselines.

**Strengths:**

- Robust and reliable CATE model evaluation is an important and often widely overlooked challenge in causal inference
- The paper is well-structured and easy to follow
- The authors provide proper theoretical derivation and extensive experimental validation for their method.

**Weaknesses:**

- The stated robustness against unobserved confounding sounds like a silver bullet. Here, the authors, provide experiments, but should more clearly discuss in theory why and to which degree of unobserved confounding their metric is more robust compared to others. Especially, since for selecting the optimal ambiguity radius they assume unconfoundedness. Hence, it is not clear if or why the metric is still robust, even though the experiments seem to support that.
- The experimental results seem to be untransparent. The experimental setup for the baselines should be described in more detail, e.g. in the Appendix. It is not clear, if the models used for the baseline metrics were properly tuned. In particular, there is high standard variation in the Regret in Table 1, and large discrepancy compared to the observed performance in ranking in Table 2. Here, the setup should be described in full detail and possible reasons for this discrepancy should be discussed more clearly.

**Questions:**

- For selecting the ambiguity radius in Proposition 3.6, unconfoundedness is assumed. So why should the metric be still robust against unobserved confounding? Was the ambiguity radius in setting C selected differently?
- Why is the ranking performance clearly worse than the average regret? The high standard deviation in the regret could imply that single runs with bad performance skew the evaluation which could be due to insufficient tuning of the used baseline models for the metrics. Clarification here would be appreciated.
- What exactly do the authors mean in the limitations when they state “considering the ambiguity set constructed with other divergence such as Wasserstein may contain more diverse distributions”? What does more diverse mean here and what are possible shortcomings of KL divergence in this setting?

**Limitations:**

Limitations are mentioned sufficiently.

---

> ### Author Rebuttal · Authors · 2024-08-06
>
> Dear Reviewer e7g2,
>
> We greatly appreciate your thoughtful feedback and insightful comments. Thank you for your recognition of our theoretical and experimental analysis! We will address each of your comments below.
>
> **Q1.** Should more clearly discuss why and to which degree of unobserved confounding their metric is more robust compared to others.
>
> **R1.** Regarding "why": Please kindly refer to GR1 of the top-page rebuttal.
>
> Regarding "to what degree", the level of robustness depends on the choice of the ambiguity radius $\epsilon$. Theoretically, a larger $\epsilon$ should guarantee the DRM-selected estimator to be more robust to hidden confounding, as it allows for a broader range of possible counterfactual distributions in the ambiguity set. However, setting $\epsilon$ too large can result in overly conservative estimator selection (similar to the well-known accuracy-robustness tradeoff). Therefore, as lines 216-220 explain, determining $\epsilon$ involves a careful balance between ensuring the counterfactual distribution is contained in the ambiguity set (i.e., robustness) and maintaining a tight upper bound (i.e., accuracy).
>
> **2.** In Proposition 3.6, unconfoundedness is assumed. Why should the metric still be robust against unobserved confounding? Was the ambiguity radius in setting C selected differently?
>
> **R2.** As discussed above, the DRM-selected estimators should be robust against unobserved confounders provided that $\epsilon$ is set appropriately. However, determining the proper value of $\epsilon$ remains an open challenge in the distributionally robust optimization literature [29, 46, 39, 41, 63]. In our work, Proposition 3.6 can guide us to set $\epsilon$ when unconfounded. When unobserved confounders are present, we can set $\epsilon$ to a larger value than the one guided by Proposition 3.6. The reason is:
>
> $D_{KL}(P_C||P_T)$
>
> $=\int_{\mathcal{X}}\int_{\mathcal{Y}^0}\int_{\mathcal{Y}^1}p(y^0,y^1|x,T=0)p(x|T=0) \log \frac{p(y^0,y^1|x,T=0)p(x|T=0)}{p(y^0,y^1|x,T=1)p(x|T=1)}dy^1dy^0dx$
>
> $=D_{KL}(P^C_X || P^T_X)+\int_{\mathcal{X}}\int_{\mathcal{Y}^0}\int_{\mathcal{Y}^1}p(y^0,y^1|x,T=0)p(x|T=0) \log \frac{p(y^0,y^1|x,T=0)}{p(y^0,y^1|x,T=1)}dy^1dy^0dx$
>
> $>D_{KL}(P^C_X || P^T_X)$
>
> In setting C, yes, the ambiguity radius is set differently. We set $\epsilon_1=D_{KL}(P^C_X || P^T_X)+5$ and $\epsilon_0=D_{KL}(P^T_X || P^C_X)+5$. This ensures that the ambiguity set is sufficiently large to contain the possible uncertain counterfactual distributions. Based on your insightful question, we will further emphasize the ambiguity radius for unmeasured confounding in the revised paper.
>
> **Q3.** The high variance for baselines in regret and worse ranking performance of DRM.
>
> **R3.** The high variance phenomenon is discussed in GR2 of the top-page rebuttal and the discussion will be added in the revised paper. Regarding the ranking performance of DRM, we have provided some explanation in lines 310-317 of the original paper. Now we provide some additional discussion.
>
> * First, high rank correlation does not necessarily imply low regret. E.g., in Table 3 of the rebuttal pdf, we have 24 CATE estimators with true performance rank in ascending order: [1, 2, 3, ..., 24]. A plug-X selector may give a surrogate PEHE value, 11.59, then it selects the 6th estimator and gives a rank order [6, 7, 8, 5, 4, 3, 9, 10, 11, 2, 1, 12, 13, 14, ..., 24]. The Spearman correlation between the true rank and Plug-X rank is 0.89. However, the regret of the selected 6th-ranked estimator is 11.63, much larger than 2.00 produced by the top-ranked estimator.
>
> * Second, the DRM approach is designed to select estimators based on their distributionally robust (worst-case) performance. Consider that a coach selects athletes to attend the Olympics based on their scores over 100 games. The first athlete consistently scores between 90-95, with an average of 93, while the second athlete's scores fluctuate between 85-96, with an average of 94. The coach may prefer to select the first athlete, as the worst-case performance is much higher, even though the second athlete has a higher average score (94) and best score (96).
>
> **Q4.** Should discuss baseline setups: whether models for baseline metrics were properly tuned.
>
> **R4.** The baseline setups and hyperparameter tuning method in our paper are in line with [16]. Details are provided in GR3 of the top-page rebuttal. We will add these in the revised paper.
>
> **Q5.** What does “more diverse mean” and what are possible shortcomings of KL divergence?
>
> **R5.** By "more diverse", we mean the ability to consider a broader range of distribution types and supports within the ambiguity set. The KL-divergence defined in Eqn (6) is a non-symmetric measure and requires P and Q to have the same distribution type and support.
>
> The limitations of the KL divergence:
> * Non-symmetry: The lack of symmetry means the KL divergence does not satisfy the triangle inequality, which can limit its applicability in certain areas, such as domain adaptation and transfer learning. Additionally, it makes the distributional distance hard to interpret.
> * Requirement for the same distribution type and support: This restriction can reduce the flexibility of the KL divergence-based ambiguity set. E.g., it prevents comparisons between continuous and discrete distributions.
>
> Therefore, We recognize that using a Wasserstein-based ambiguity set would be a promising research direction. It can be used to compare distributions of different types and supports, capturing a wider range of possible counterfactual distributions. However, as mentioned in the paper, the dual problem of the Wasserstein version involves an intractable "sup" problem, which presents additional challenges that require further investigation.
>
> Thank you again for your time and effort in providing these constructive insights that help improve the clarity and rigor of our work! We are very welcome to any additional feedback you may have.

---

> > ### Comment · Reviewer_e7g2 · 2024-08-13
> >
> > Thank you for your rebuttal, it addressed most of my concerns.
> >
> > In general, I think that the writing of the paper could be improved at some points, e.g., less subjective statements and more clarity and transparency regarding the implementation.
> > Therefore, please include the additional explanations regarding the empirical evaluation, especially the exact tuning settings and the considered ambiguity radius, in the paper or appendix.
> >
> > Further, I would strongly encourage the authors to include an additional sensitivity study over varying values of the ambiguity radius under unobserved confounding and recommendations how to select the ambiguity radius in practice.
> >
> > However, overall I think this is still an interesting and valuable paper. Hence, I will increase my score by 1.

---

> ### Author Response · Authors · 2024-08-13
> **Response to Reviewer e7g2**
>
> Dear Reviewer e7g2,
>
> Thank you for your thoughtful reply. We are delighted our rebuttal has addressed most of your concerns, and we greatly appreciate your suggestions and will implement them as follows:
>
> Regarding the experimental aspects, we will include more details, such as settings, model training procedures, hyperparameters, and the choice of ambiguous radius under the hidden confounder setting. We believe these additional materials will enhance the clarity and transparency of our work.
>
> As for your suggestion concerning the ambiguity radius, we will incorporate a sensitivity analysis for the proposed DRM under the hidden confounders setting. We believe this analysis will provide valuable guidance for practitioners and researchers in selecting appropriate ambiguity radius when unobserved confounders exist.
>
> We will carefully revise our paper as we respond in rebuttal. Once again, thank you for taking the time to review our paper, and we appreciate your recognition of our work!
>
> Best regards,
>
> Authors 4584

---

### Official Review · Reviewer_T6Zv · 2024-07-21

**Soundness:** 2
**Presentation:** 3
**Contribution:** 2
**Rating:** 6
**Confidence:** 4

**Summary:**

The paper proposes a new model selection method for choosing an estimator of the conditional average treatment effect (CATE), namely, a Distributional Robust Metric (DRM). The proposed method, DRM, is nuisance-free: It does not require an additional estimation of the nuisance functions, unlike the majority of existing approaches. The DRM splits a precision in the estimation of the heterogeneous effects (PEHE) into two parts: the first term can be estimated nuisance-free and the second term is upper-bounded using the KL-divergence ambiguity set. Thus, the proposed method can only utilize the observational data.  Furthermore, the authors provided an estimation procedure for the DRM and the finite-sample convergence rates. Finally, multiple synthetic experiments were provided to prove the effectiveness of the method in comparison to the other existing approaches.

**Strengths:**

The paper studies an important and heavily discussed problem in causal inference, CATE model selection. I find the idea of using the upper bound on the PEHE based on the distributional ambiguity set original and novel. The paper is well-structured and easy to follow.

**Weaknesses:**

Two major weaknesses of the paper are, in my opinion, the following:
1. A lack of understanding of the asymptotic behaviour of the DRM. Specifically, when the data size grows, the upper bound on the term (b) of Eq. (5) will still stay the same, as the epsilon-ball around $P_C$ or $P_T$ stays constant. Hence, there will be always a gap between the ground-truth PEHE and the DRM from Eq. (14). This renders the DRM not consistent. I encourage the authors to provide a discussion about this issue and some experiments with the varying data sizes, where they compare the DRM with other (consistent) nuisance-free selectors, e.g., nearest-neighbours matching.
2. It was not directly clear to me, why the optimal $\epsilon^*$ is a forward KL-divergence but not, e.g., a reverse KL-divergence between treated and untreated covariates distributions. Note, that KL-divergence is not symmetric. Does it make sense to use the symmetric distributional distances?

There are also several minor suggestions for improvement:
1. The hidden confounding setting needs to be considered more carefully. This setting requires different approaches, e.g., partial identification or sensitivity models, and, thus, fitting/selecting point CATE models is inadequate in this case.
2. Some baselines were not mentioned in the related work or the experiments, e.g., U-learner [1].

I am open towards raising my score if the authors address the above-mentioned issues during the rebuttal.

References:
- [1] Fisher, Aaron. "The Connection Between R-Learning and Inverse-Variance Weighting for Estimation of Heterogeneous Treatment Effects." arXiv preprint arXiv:2307.09700 (2023).

**Questions:**

- What ‘uncertainty in PEHE’ is meant in lines 164-165?

**Limitations:**

The main limitation of the method, i.e., the lack of consistency, was not discussed by the authors.

---

> ### Author Rebuttal · Authors · 2024-08-06
>
> Dear Reviewer T6Zv,
>
> We are grateful for your thorough summary and greatly appreciate your recognition of the motivation and novelty underlying our proposed method. Thank you for your time and effort in providing constructive and helpful feedback. We will respond to each of your comments below.
>
> **Q1.** DRM is not consistent with ground-truth PEHE. Better discuss this issue and do some experiments with the varying data sizes, comparing with other (consistent) nuisance-free selectors, e.g., nearest-neighbours matching.
>
> **R1:** Thank you for this suggestion. First, we would like to emphasize that we did not claim the proposed DRM metric converges to the ground-truth PEHE. Our theoretical results only state that $\hat{\mathcal{V}}^t$ should converge to the $\mathcal{V}^t$ at a rate of $1/\sqrt{n}$. Consequently, together with the law of large numbers for other terms in $\mathcal{R}^{DRM}(\hat{\tau})$ in Eqn. (14) should converge to $\mathcal{V}_{PEHE}(\hat{\tau})$ in Eqn. (9) at a rate of $1/\sqrt{n}$. Additionally, we believe it is not a weakness that DRM is not consistent with PEHE for the following two reasons:
>
> * By definition, DRM measures the robustness of PEHE w.r.t. $\hat{\tau}$. We can never observe the ground-truth PEHE due to the uncertainty incurred by selection bias and hidden confounders (as discussed in GR_1 of the top-page rebuttal). Therefore, instead of pursuing the unavailable PEHE, we aim to quantify its uncertainty using a distributionally robust value of PEHE: If some $\hat{\tau}$ has a small $\mathcal{R}^{DRM}(\hat{\tau})$, then $\hat{\tau}$ is robust to the PEHE uncertainty caused by the uncertainty in counterfactual distribution.
> * To the best of our knowledge, no existing nuisance-free estimator/selector (e.g., the mentioned matching) is assured to be a consistent estimator of the ground-truth CATE/PEHE, but some nuisance-free estimators are consistent estimators of ATE (not CATE) given additional assumptions. E.g., (Abadie & Imbens, 2006) show the matching estimator converges to ATE at a rate slower than $1/\sqrt{n}$, given the assumption that the density of covariates is bounded.
>
> Per your suggestion, we have compared DRM and nearest-neighbor matching with varying sample sizes in Table 2 of the attached rebuttal pdf reports the details and results. The findings show that DRM and matching exhibit a decreasing trend in regret with increasing sample size. However, when the sample size exceeds 10,000, the average regret for matching fluctuates between 6.5 and 7, while the average regret for DRM becomes steady. These results suggest that neither DRM nor matching converges to the ground-truth PEHE because a consistent metric should have a regret that tends to zero. Nevertheless, DRM demonstrates more stable and effective performance than matching as the sample size varies.
>
> Ref: (Abadie & Imbens, 2006) Large sample properties of matching estimators for average treatment effects, Econometrica, 2006
>
> **Q2.** Why the $\epsilon^*$ is determined as $D_{KL}(P_C||P_T)$ instead of $D_{KL}(P_T||P_C)$? Does it make sense to use the symmetric distributional distances?
>
> **R2.** Indeed, we set $\epsilon_1^*=D_{KL}(P^C_X||P^T_X)$ and $\epsilon_0^*=D_{KL}(P^T_X||P^C_X)$ in the original paper. In lines 222-237, we intended to explain the choice of $\epsilon^*$ with $\epsilon_1^*=D_{KL}(P^C_X||P^T_X)$ as an example, but forgot to explain $\epsilon_0^*$, which led to a misleading presentation. We are grateful for your pointing this out, and we will revise the incomplete expression to make it clearer.
>
> Regarding your second question, we believe using symmetric distances would be more meaningful. The non-symmetric nature of KL divergence makes it fail to satisfy the triangle inequality and can make the divergence values difficult to interpret and compare between distributions. This restricts its practical use in fields like domain adaptation and transfer learning. Exploring the use of symmetric divergences, such as the Wasserstein distance, would be very useful in DRM (as discussed in line 343). However, solving the distributionally robust optimization problem with a Wasserstein-based ambiguity set can be computationally challenging due to the intractable dual problem involving an infinite "sup" objective. Therefore, considering alternative symmetric divergences that may provide a more tractable optimization framework for DRM would be a promising direction.
>
> Minor suggestions:
>
> **S1.** More baseline CATE estimators aiming at the hidden confounding setting should be considered.
>
> **R1.** At the current stage, one key focus is developing a CATE selection method that can robustly handle the distributional shifts. Therefore, the paper is mainly around “selection” rather than “estimation”. However, we strongly agree that considering more CATE estimators aiming at hidden confounder issues would provide more insights for testing different selectors under the hidden confounder setting, which will be further investigated. We will mention this interesting suggestion and leave it for future research.
>
> **S2.** U-learner can be included as a baseline CATE estimator.
>
> **R2.** We have incorporated the U-learner [1] into our experimental evaluation, and the updated results are reported in Table 1 of the top-page rebuttal pdf. We will add a discussion of the U-learner and incorporate the updated experimental results in the final version of the paper.
>
> **Q.** What 'uncertainty in PEHE' is meant in lines 164-165?
>
> **R.** We hope GR_1 of the top-page rebuttal has addressed this question well, and we will add these details in the Appendix to provide clearer explanations for the proposed method.
>
> Finally, thank you again for providing a thorough review and constructive suggestions! Your comments are valuable in helping us improve our work, and we will carefully incorporate your suggestions when finalizing the paper. We are very welcome any further questions or comments you may have.

---

> ### Comment · Reviewer_T6Zv · 2024-08-12
>
> Thank you for clarifying most of the concerns.
>
> Still, it seems to me that the $k$-nearest neighbours estimator would consistently estimate both of the potential outcomes surfaces, $\mathbb{E}(Y \mid X =x, T = t)$, if $k$ is chosen in a data-driven way. I wonder how the authors chose the hyperparameter $k$ for the new synthetic experiments.

---

> ### Author Response · Authors · 2024-08-12
> **Response to Reviewer T6Zv**
>
> Dear Reviewer T6Zv,
>
> Thank you for your reply. We are pleased to hear that we have successfully addressed most of your concerns!
>
> In both our original experiment (e.g., Table 1 in original paper) and new experiments, we set $k=1$ for the k-nearest neighbours estimator, which directly follows the code and strategy of the k-nearest neighbours selection metric in [16] to maintain consistency with literature.
>
> Aditionally, if possible, we would appreciate any additional information or references about the claim that "the k-nearest neighbours estimator would consistently estimate both of the potential outcomes surfaces if k is chosen in a data-driven way." To the best of our knowledge, we have not found any evidence supporting this conclusion. We sincerely hope to address this and improve the current comparison. Your suggestions would be valuable in improving the kNN baseline, and further strengthen the experiment quality of our paper.
>
> Thank you once again for your time in reviewing our paper and responding to our rebuttal. We really look forward to your further reply!
>
>
> Best regards,
>
> Authors 4584

---

> > ### Comment · Reviewer_T6Zv · 2024-08-12
> >
> > Thank you for the additional details.
> >
> > Regarding the consistency of the knn estimator: What about the following work?
> > -  Devroye, Luc, et al. "On the strong universal consistency of nearest neighbor regression function estimates." The annals of Statistics 22.3 (1994): 1371-1385.

---

> ### Author Response · Authors · 2024-08-13
> **Response to Reviewer T6Zv**
>
> Dear Reviewer T6Zv,
>
> Thank you for sharing this reference and giving us a chance to discuss it with you. We strongly agree with you that the kNN regressor can be a consistent estimator of target functions in some machine learning contexts. **However, such consistency property may not necessarily hold when fitting potential outcome surfaces due to the distribution shift between factual and counterfactual distributions.**
>
> The paper you mentioned and Chapters 6 and 11 of [1] prove the consistency of the kNN regression estimator under certain conditions, such as $k/n \rightarrow 0$. However, note that **the consistency holds with the assumption that data are independent and identically distributed (i.i.d.)**, as discussed in the Introduction of [1]. Though the i.i.d. assumption is common in most machine learning scenarios, it doesn't necessarily hold when estimating potential outcome surfaces $\mu_t$ for $t \in $ {0, 1}. Please allow us to explain in detail below.
>
> In the Rubin Causal Model framework, observational (factual) data are i.i.d. tuples $(x_i,t_i,y_i)_{i=1}^{n}$, following the factual distribution $P^{F}:=P(X,T,Y)$. For each pair $(x_i,t_i)$, there also exists an unobserved counterfactual outcome $y_i^{CF}$. The unobserved (counterfactual) data are i.i.d. tuples $(x_i,t_i,y_i^{CF}) _{i=1}^{n}$, following the counterfactual distribution $P^{CF}:=P(X,T,Y^{CF})$. As explained in Section 3.1 and GR1 of our top-page rebuttal, **$P^{F}$ is not identical to $P^{CF}$ in general**. Consequently, a model trained on $P^{F}$ may not predict well on $P^{CF}$.
>
> Let's consider estimating $\mu_0(x)$ for further explanations. To infer the potential outcome $Y^0$ for treated ($T=1$) samples, the kNN regressor first approximates $\hat{\mu}_ {0}(X)$ using controlled samples $(X_i,Y^0_i) _ {i=1}^{n_ {control} }$, then uses $\hat{\mu} _ {0}(X)$ to predict $Y^0$ for treated samples, with the predicted samples being $(X_i, \hat{Y}^0_i) _ {i=1}^{n_ {treat} }$. However, $\hat{\mu} _ {0}(X)$ does not necessarily generalize well on $Y^0$ predictions for treated samples, because the source training data (controlled samples) do not have the same distribution as the target prediction data (treated samples), meaning that there exists a distribution shift problem $P(X,Y^0|T=0) \neq P(X,Y^0|T=1)$. Such a distribution shift (discussed in Section 3.1 and GR1 of our top-page rebuttal) renders kNN-based learner $\hat{\mu}_{0}(X)$ to be a consistent estimator of $\mu_0(x)$.
>
> Therefore, the above analysis demonstrates that, if kNN-based learner $\hat{\mu}_ {0}(X)$ would be a consistent estimator of $\mu_ {0}(X)$, one of the following two conditions must be met:
>
> * The factual distribution for controlled samples, $P(X_i,Y^0_i|T_i=0)$, should be identical to the counterfactual distribution for treated samples, $P(X_i,Y^0_i|T_i=1)$. This necessitates the absence of distribution shift between factual and counterfactual distributions, which in turn requires two key conditions: the unconfoundedness and access to infinite data from Randomized Controlled Trials (RCTs).
>
> * The training samples should be $(X_i,Y^0_i)$ (including the whole samples) instead of $(X_i,Y^0_i)|T_i=0$ (only including controlled samples). This requires counterfactual knowledge, because the potential outcome $Y^0_i$ remains unknown for treated ($T_i=1$) data.
>
> We hope this clarifies our perspective on the consistency of kNN in causal inference contexts. Note that the above analysis and conditions for consistency property are not exclusive to kNN but also extend to other machine learning methods employed in estimating potential outcomes or CATE.  We will further incorporate the above discussions in Appendix, as they further underscore the importance and significance of our method in CATE estimator selection. The DRM method offers distinct advantages compared with previous selection metrics: it does not require counterfactual data and is capable of selecting estimators that are robust to the distribution shift between factual and counterfactual samples.
>
> Thank you again for your active engagement and effort in helping us improve the paper! We are very welcome any further thoughts or questions you may have.
>
> Reference: [1] Devroye, Luc, et al. A Probabilistic Theory of Pattern Recognition. Stochastic Modelling and Applied Probability.
>
> Best regards,
>
> Authors 4584

---

> > ### Comment · Reviewer_T6Zv · 2024-08-13
> >
> > Thank you for the quick response!
> >
> > Yet, I still think that the k-NN is a consistent estimator (an instantiation of the T-learner). The reason is that the distribution shift (or selection bias) only matters for CATE estimation in the low-sample regime [1]. In large-sample regimes, any possible universal function approximator can serve as a consistent estimator of $\mu_0(x)$ and $\mu_1(x)$. Thus, the distribution shift does not hinder the consistency of estimation.
> >
> > Nevertheless, I consider this paper very interesting and thought-provoking. Therefore, I will raise my score.
> >
> > P.S. Here is something that I realized after submitting the review for this paper (therefore the following does not influence my valuation of the current work). There was a recent concurrent paper from ICML 2024, which tackles the same problem of CATE model selection/validation [2]. There, the authors used a _total variation distance_ to bound the counterfactual term of the PEHE risk. Also, they compared their work with the _integral probability metrics_ bounds from [3]. In the final version of the manuscript, I encourage the authors of this work to incorporate the discussion on the above-mentioned works [2, 3] (with alternatives to KL-divergence).
> >
> > References:
> > - [1]  Alaa, Ahmed, and Mihaela Schaar. "Limits of estimating heterogeneous treatment effects: Guidelines for practical algorithm design." International Conference on Machine Learning. PMLR, 2018.
> > - [2] Csillag, Daniel, Claudio José Struchiner, and Guilherme Tegoni Goedert. "Generalization Bounds for Causal Regression: Insights, Guarantees and Sensitivity Analysis." arXiv preprint arXiv:2405.09516 (2024).
> > - [3] Uri Shalit, Fredrik D Johansson, and David Sontag. Estimating individual treatment effect generalization bounds and algorithms. In International conference on machine learning, pages 3076–3085. PMLR, 2017.514

---

> ### Author Response · Authors · 2024-08-13
> **Response to Reviewer T6Zv**
>
> Dear Reviewer T6Zv,
>
> Thank you for your thorough and insightful suggestions! We find that the paper [1] you mentioned is very useful for both CATE estimation and model selection. Their findings, which reveal that the minimax rate of PEHE depends on smoothness and sparsity rather than selection bias, align well with the key investigation in [16] regarding how CATE complexity affects model selection metrics. This has provided us with new insights: How can we select estimators that achieve the optimal PEHE minimax rates? We believe this is a very interesting and important avenue, and it may be closely related to the smoothness and sparsity of CATE and response surfaces.
>
> We are also grateful for your second suggestion. The Integral Probability Metric (IPM), e.g., the Wasserstein distance used in [3], was briefly discussed in line 343 of our original paper. Together with your original question about "symmetric distribution discrepancy" and our rebuttal R2, we will incorporate more detailed discussions on the possible usage of Wasserstein distance in the proposed DRM method. Regarding the paper [2] you mentioned, we find their proposed upper bound for potential outcome error very interesting. They claim that the bound will be tight given a proper selection of the hyperparameter λ. Such a tightness property suggests a potential for f-divergence in the DRM method. In light of this, we intend to conduct a comparative analysis of the advantages and disadvantages of using IPM (such as Wasserstein or MMD) and f-divergence versus KL-divergence.
>
> Finally, we want to express our gratitude again for your quick replies and insightful discussions. The suggested references are valuable and insightful for our future research. Thank you for your time, expertise, and comments on helping us improve the study!
>
> Best regards,
>
> Authors of 4584

---

### Author Rebuttal · Authors · 2024-08-05

Dear Reviewers,

We are grateful for your comments and suggestions, which are very helpful in improving our paper. Here are some general responses (GR) that might be useful for individual questions. In the separate rebuttal, we may remind you to refer to this general response. Thank you for your time and effort during the review process!

**GR1. DRM is able to select CATE estimators that are robust to the uncertainty in PEHE caused by selection bias and unobserved confounders.**

In Sections 3.1 and 4.1, we have presented theoretical explanations for why the DRM metric can measure a CATE estimator's robustness against selection bias and unobserved confounding. To address any potential questions from reviewers on this topic, we will provide a more specific explanation below, and these details will be further added to the Appendix.

In causal inference, all the CATE estimators are constructed using observational factual data. However, we can never know how reliable the CATE estimator is due to the unavailable Oracle PEHE in Eqn. (1). As shown in Eqn. (5),  the PEHE is equal to two $\hat{\tau}$-dependent terms, $\mathbb{E}[\hat{\tau}(X)Y^t|T=t]$ and $\mathbb{E}[\hat{\tau}(X)Y^t|T=1-t]$. Unfortunately, $\mathbb{E}[\hat{\tau}(X)Y^t|T=1-t]$ is  in practice. This is because we can only observe the factual distribution $P^F = P(X, Y^t | T=t)$, but not the counterfactual distribution $P^{CF} = P(X, Y^t | T=1-t)$. The unobserved counterfactual distribution can be regarded as an uncertain distribution varying around the observed and certain factual distribution $P^{F}$. If we could assume a "God's perspective" and observe $P^{CF}$ directly, the counterfactual distribution would be certain - like a quantum world! Such an uncertainty in $P^{CF}$ results in the uncertainty in PEHE. In the following, we analyze the source of this uncertainty by examining the relationship between the uncertain distribution $P^{CF}$ and the certain distribution $P^F$:

$P(X,Y^t|T=1-t)=P(X,Y^t|T=t) \frac{P(Y^t|T=1-t, X)}{P(Y^t|T=t, X)} \frac{P(X|T=1-t)}{P(X|T=t)}$.

Therefore, the unobservable distribution $P(X, Y^t | T=1-t)$ can be expressed as the observable distribution $P(X, Y^t | T=t)$ multiplied by the term $\frac{P(Y^t | T=1-t, X)}{P(Y^t | T=t, X)} \frac{P(X | T=1-t)}{P(X | T=t)}$. In other words, the ratio $\frac{p(y^t | T=1-t, x)}{p(y^t | T=t, x)} \frac{p(x | T=1-t)}{p(x | T=t)}$ controls the discrepancy between the factual distribution $P^F$ and the counterfactual distribution $P^{CF}$. Note that if there are no unmeasured confounders, then we have $p(y^t | T=1-t, x) = p(y^t | T=t, x)$; and if there is no selection bias (covariate shift), then we have $p(x | T=1-t) = p(x | T=t)$. Now we understand that the root cause of the discrepancy between $P^F$ and $P^{CF}$ (or between $\mathbb{E}[\tau(X)Y^t | T=1-t]$ and $\mathbb{E}[\tau(X)Y^t | T=t]$) lies in the presence of unobserved confounders and selection bias. In Section 4.1, the uncertainty caused by potential unobserved confounders and selection bias in PEHE can be further measured as the distributionally robust values $\mathcal{V}^t$. Then the PEHE w.r.t. the CATE estimator $\hat{\tau}$ will be at most $\mathcal{V}^t$, which reflects the distributional robustness of $\hat{\tau}$.


**GR2. The high variance regret in some baseline selectors.**

This phenomenon is primarily due to the wide range of PEHE performances produced by a total of 24 CATE estimators. For example, in setting A ($\rho=0.1$), the average PEHE ranged from 2.00 for the best estimator to 141.01 for the worst estimator (we will present the PEHE performances of all 24 CATE estimators in the Appendix). This significantly large gap between the good and bad estimators leads to high variance of some baseline selectors - if a selector is able to consistently select either good or bad estimators, the variance would not be as pronounced. Therefore, it is important to further analyze the performance of different selectors in each experiment.

To achieve this, we sorted all 24 estimators in ascending order of PEHE and determined the rank of the estimator selected by each selector in each of the 100 experiments. In lines 318-326 and Figure 1 of Appendix C.1, we find that many baseline methods tend to select CATE estimators ranked across different percentile ranges, resulting in a high variance in 100 selections. In contrast, the DRM selector is able to consistently select higher-ranked (i.e., better performing in PEHE) estimators, while mitigating the risk of selecting lower-ranked estimators in most cases. This not only confirms the robustness strength of the DRM but also helps explain its lower variance observed in Table 1.

**GR3. Hyperparameters tuning for nuisance training.**

We have tuned hyperparameters for each model whenever there is a model training process. Specifically, we used RidgeCV and LogisticRegressionCV for the linear regression and logistic regression models, respectively. We also employed GridSearchCV for the SVM and Random Forest models. The number of cross-validation folds was set to 3. The specific hyperparameter ranges we searched are as follows:

Ridge regression: $\alpha \in$ \{ 0.01, 0.1, 1.0, 10.0, 100.0 \}.

Logistic regression: $C \in$ \{0.01, 0.1, 1, 10\}.

SVM: Kernel $\in$ \{Sigmoid, RBF\}, $C \in$ \{1, 5\}.

RF and XGBoost: Max depth $\in$ \{1, 3, 6\}, n\_estimator $\in$ \{20, 100\}.

---

### Decision · Program_Chairs · 2024-09-25

**Decision:**

Accept (poster)

**Comment:**

This paper proposes a distributionally-robust method for selecting CATE estimators. Reviewers were generally very positive about the paper's contributions and emphasized many of its strength. Some limitations (e.g., the non-symmetric nature of D_KL) were discussed in length in a way that seems to have both grounded the author's choices and results, as well as improved the paper. Others should be implemented into the next revision (e.g., details and clarity regarding experimental setup and evaluation, inaccuracies and redundancies of some claims). One point that was brought up multiple times is the extended use of subjective statements - these should be reduced considerably (or removed altogether) in the final version. Nonetheless, overall this is a good paper that is likely to advance the current state of research on CATE model selection and promote the importance of robust models in this literature.